# Gpr54 deletion accelerates hair cycle and hair regeneration

Weili Xia[1,2,3], Caibing Wang [ID][1,3], Biao Guo [ID][1,3], Zexin Tang[1], Xiyun Ye [ID][1,4✉] & Yongyan Dang [ID][1,4✉]

## Abstract

GPR54, or KiSS-1R (Kisspeptin receptor), is key in puberty initiation and tumor metastasis prevention, but its role on hair follicles remains unclear. Our study shows that Gpr54 knockout (KO) accelerates hair cycle, synchronized hair regeneration and transplanted hair growth in mice. In Gpr54 KO mice, DPC (dermal papilla cell) activity is enhanced, with elevated expression of Wnts, VEGF, and IGF-1, which stimulate HFSCs. Gpr54 deletion also raises the number of CD34+ and Lgr5+ HFSCs. The Gpr54 inhibitor, kisspeptin234, promotes hair shaft growth in cultured mouse hair follicles and boosts synchronized hair regeneration in vivo. Mechanistically, Gpr54 deletion suppresses NFATC3 expression in DPCs and HFSCs, and decreases levels of SFRP1, a Wnt inhibitor. It also activates the Wnt/β-catenin pathway, promoting β-catenin nuclear localization and upregulating target genes such as Lef1 and ALP. Our findings suggest that Gpr54 deletion may accelerate the hair cycle and promote hair regeneration in mice by regulating the NAFTc3-SFRP1-Wnt signaling pathway. These findings suggest that Gpr54 could be a possible target for future hair loss treatments.

**Keywords** Gpr54; Hair Cycle; NFATc3; SFRP1; Wnt/β-catenin
**Subject Categories** Signal Transduction; Stem Cells & Regenerative Medicine

## Introduction

The normal human hair follicle (HF) undergoes a continuous cycle consisting of the growth phase (anagen), transitional phase (catagen), and resting phase (telogen) (Oh et al, 2016). The interactions between HFSCs and DPCs play a significant role in the regulation of hair cycling. HFSCs rely on communication with their microenvironment, known as the stem cell niche, to receive external cues that regulate their activity. During the anagen phase of the hair growth cycle, DPCs can secrete growth factors such as Wnt, IGF-1, FGF, and Noggin to preserve HF-inducive capacity (Kiso et al, 2015; Trüeb, 2018; Zhou et al, 2016). Once activated, HFSCs can undergo proliferation, self-renewal, and differentiation

to replenish damaged or lost cells within hair follicles, thereby initiating the anagen phase. Hair loss, also called alopecia, is a disorder caused by an interruption in the cycle of hair regeneration. Alopecia, commonly associated with significant psychological burden for individuals, currently has limited treatment options available today. Although hair transplantation has been widely utilized, the transplanted hairs are not sustained in the long term. In addition, only a limited number of drugs are available to promote hair regeneration, and these often come with side effects. This is mainly due to the intricate molecular mechanisms involved in hair regeneration, making it difficult to identify effective molecular targets for drug development. Therefore, there is an urgent need to explore alternative therapeutic solutions that can generate functional hair follicles.

GPR54, also referred to as KiSS-1R (Kisspeptin receptor), is widely expressed in the human organs such as hypothalamus, testis, spleen, and lymph nodes (Zhu et al, 2022). KiSS-1/GPR54 signaling pathway, as a key endocrine regulator, is pivotal in maintaining onset of puberty through the activation of GnRH secretion (Zhu et al, 2022). The Arg386Pro mutant of Gpr54 appears to be associated with central precocious puberty (Teles et al, 2008). It is reported that aging reduces kisspeptin receptor (GPR54) expression levels in the hypothalamus and extra-hypothalamic brain regions (Mattam et al, 2021). Recently, KiSS-1/GPR54 system was also demonstrated to act as the aggressive and metastatic inhibitor in malignant tumors (Zhu et al, 2022). In addition, GRP54 is expressed in the immune system, regulating self-tolerance and restricting antiviral innate immune responses (Huang et al, 2018; Xing et al, 2018). These researches indicate that GPR54 plays a significant role in maintaining both human physiological and pathological functions. Hair growth is regulated by endocrine and neural stimuli in addition to nutrition and age. The Hair-GEL (Gene Expression Library) online tool provides a clue that Gpr54 may be closely associated with hair function due to its high expression in DPCs (Rezza et al, 2016; Sennett et al, 2015). However, to date, there is no documented evidences regarding the function of Gpr54 on hair follicles.

In this study, we first investigated whether GPR54 affects hair cycle and hair regrowth using a GPR54 knockout animal model and further examined the underlying molecular mechanisms in various experimental systems. We discovered that Gpr54 is abundantly expressed in the DPCs and HFSCs of mouse hair follicles. The loss of the *Gpr54* gene accelerated the telogen-to-anagen transition of

[1]Institute of Biomedical Sciences and School of Life Sciences, East China Normal University, Shanghai 200241, China. [2]Shandong Mental Health Center, Shandong University, Jinan, Shandong 250014, China. [3]These authors contributed equally: Weili Xia, Caibing Wang, Biao Guo. [4]These authors jointly supervised this work: Xiyun Ye, Yongyan Dang. ✉E-mail: xyye@bio.ecnu.edu.cn; yydang@bio.ecnu.edu.cn

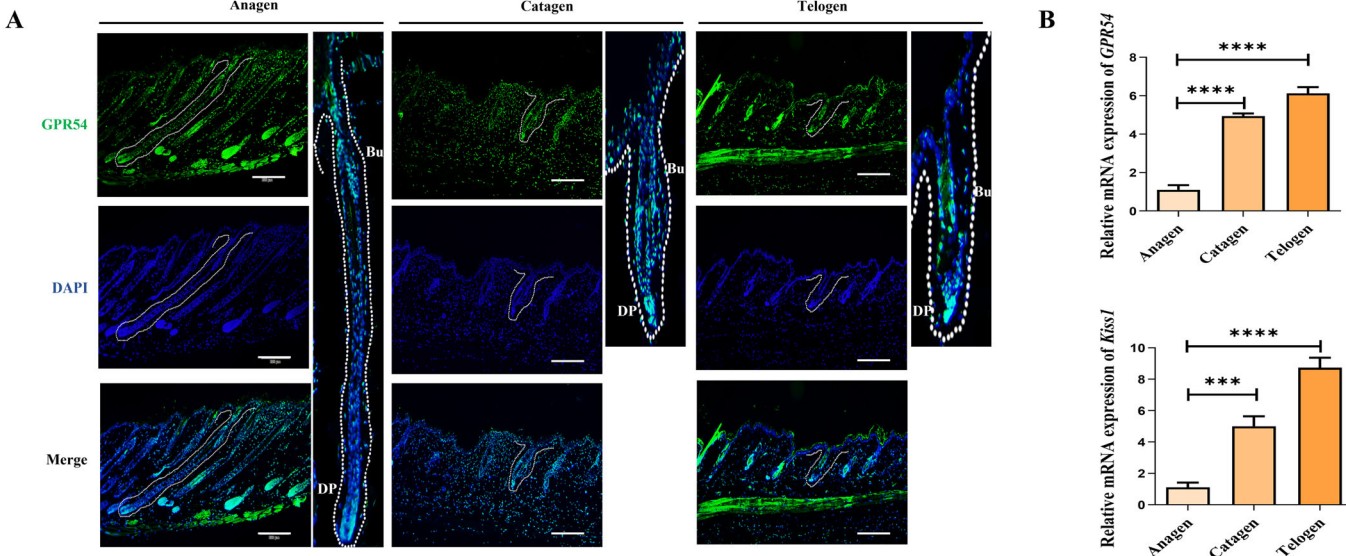

**Figure 1. The expression and localization of Gpr54 in the hair follicles.**

(A) Immunofluorescence labeling of Gpr54 in mouse hair follicles during three different hair cycle stages: anagen, catagen, and telogen. Sections were also counterstained with DAPI (blue) to visualize nuclei. DP stands for dermal papilloma, and Bu stands for bulge. Scale bar: 200 μm. (B) Q-PCR analysis of Gpr54 and kisspepeptin1 expression in the whole skin tissues of mouse from three different hair cycle stages: anagen, catagen, and telogen. Data are mean ± SD of three independent experiments. A paired t-test was performed, N = 3. For Gpr54: ****P (catagen vs. anagen) < 0.0001 and ****P (telogen vs. anagen) < 0.0001; for kiss1: ***P (catagen vs. anagen) = 0.0007 and ****P (telogen vs. anagen) < 0.0001. Source data are available online for this figure.

hair cycle, and promoted hair regeneration in mice. Moreover, Gpr54 deletion enhanced the activity of DPCs and promoted the proliferation of HFSCs in hair follicles. It appeared that Gpr54 deletion accelerated the initiation of anagen by regulating the NAFTc3-SFRP1-Wnt signaling cascades. Importantly, the Gpr54 inhibitor Kisspeptin-234 (KP234) effectively promoted the growth of hair shafts in in vitro cultured mouse hair follicles and hair regeneration in mice. Therefore, Gpr54 may be a novel molecular target for treating hair loss in the future.

# Results

## Gpr54 was dynamically expressed in postnatal hair follicles

In order to confirm if Gpr54 is involved in the regulation of hair cycle, we first analyzed Gpr54 localization in hair follicle by immunofluorescent staining. Skin samples of mouse backs were taken at anagen (day 35), catagen (day 42), and telogen (day 49) of the second hair cycle, respectively. In each stage of mouse hair cycle, Gpr54 was mainly expressed in the bulge, hair germ and dermal papilla (DP) (Fig. 1A). Meanwhile, the lack of Gpr54 staining in the hair follicles of Gpr54 knockout (KO) mice validated the specificity of GPR54 antibody (Fig. EV1A). Strong Gpr54 fluorescence staining in both the DP and HFSC regions indicates high expression of Gpr54 in these areas (Fig. 1A). Weak staining can be also observed in the outer root sheath. Q-PCR results showed that the expression of Gpr54 and its ligand kisspeptin 1 (KISS1) in the whole mouse skin tissues during catagen and telogen were higher than that in anagen skins (Fig. 1B). Together, the

pronounced expression of Gpr54 in the bulge, hair germ, and dermal papilla suggests that GPR54 may play a role in regulating the hair cycle.

## Gpr54 deficiency promoted the transition from telogen to anagen in mouse

To investigate if loss of Gpr54 affected the normal HF cycle, skin color and HE-stained sections of wild-type (WT) and knockout (KO) mice from P8-P73 were observed to evaluate the changes of hair cycle. Although both Gpr54 WT and KO mice were in the first telogen phase of the hair cycle at P19, this phase persists until day 23 in WT mice, whereas in KO mice, it only lasts until day 21. By day 23, KO mice have entered the early stage of the second hair cycle's anagen phase, while WT mice remained in the first telogen stage. By P25, GPR54 KO mouse hair follicles had progressed to the second anagen VI phase, whereas WT mice were still in the second anagen III, in both males and females (Fig. 2A). Consistently, the immunofluorescent results indicated that, at 25 days, the number of Ki67-positive cells in the hair bulb of KO mice was significantly higher than the WT mice (Fig. EV1B). In WT mice, the second telogen phase began at day 50 and continued until day 73, without transitioning into the third anagen phase of hair cycle. In contrast, KO mice had already entered the anagen phase of the third hair cycle by day 50 and remained in anagen through day 73 (Fig. 2A). Obviously, Gpr54 deletion can elongate the anagen phase of the hair growth cycle and shorten telogen (Fig. EV1C). Histological analysis also revealed a faster transition from telogen to the subsequent anagen in both male and female Gpr54 mutants (Fig. 2A). Interestingly, we found that Gpr54 did not alter the total amounts of hair follicles in all the three stages of hair cycle (Fig. EV1D,E).

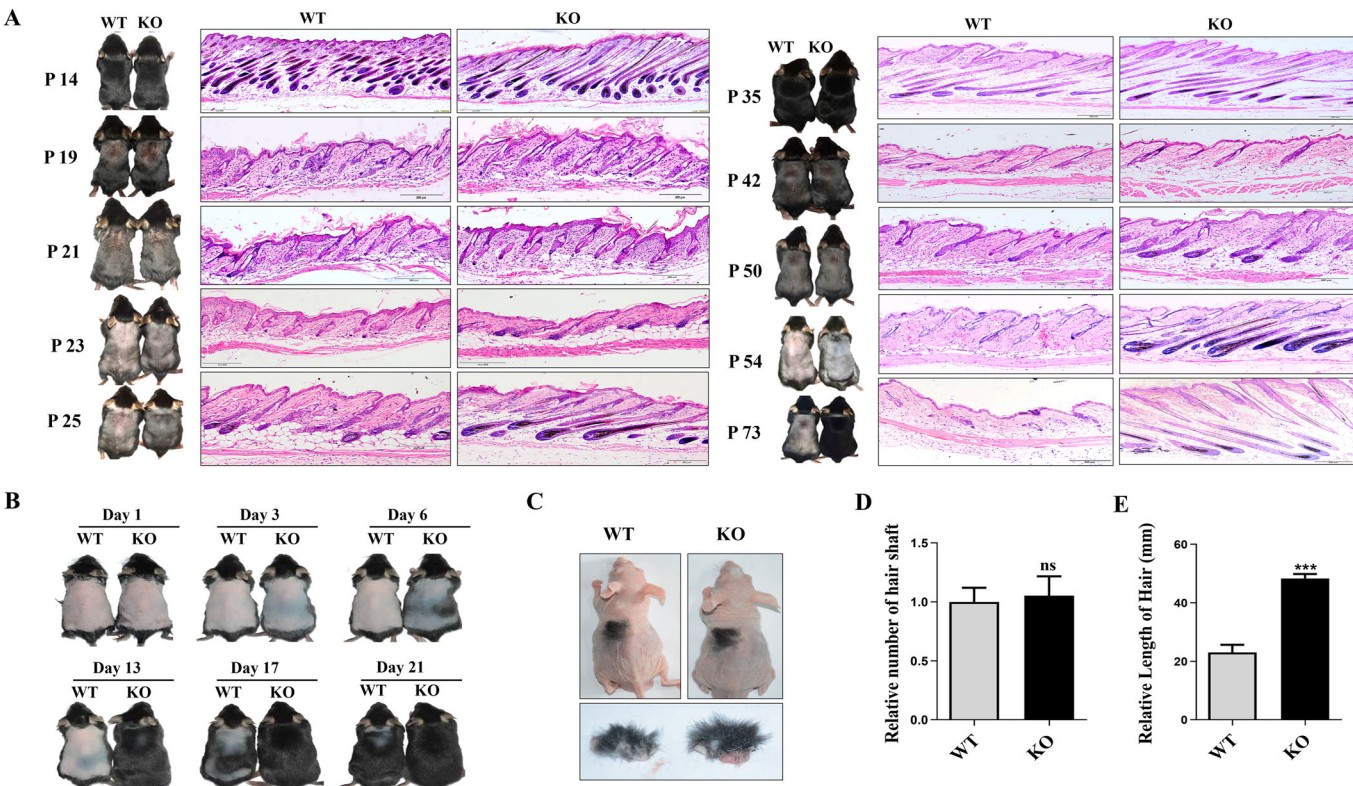

**Figure 2.** *Gpr54* knockout promoted the transition from telogen to anagen and hair regrowth.

(A) The photos (left) of hair cycle and H&E staining images (right) from skin tissue sections of *Gpr54* WT and KO mice at day 14, 19, 21, 23, 25, 35, 42, 50, 54, and 73 after birth. Scale bar: 200 μm. (B) Hair regeneration after depilation in 8-week-old *Gpr54* WT and KO littermate mice. Macroscopic images at 1, 3, 6, 13, 17, and 21 days after depilation. (C) Skins of 8 mm diameter from *Gpr54* WT and KO littermate mice at the age of 19 days were transplanted onto the skin of BABL/C nude mice. The upper shows the images of hair growth after transplantation for ten days. The lower displays the separated transplants. (D, E) The average of hair number and hair length of separated transplants from *Gpr54* WT and KO littermate mice were counted. Data are mean ± SD of three independent experiments. A paired *t*-test was performed, $N = 3$, ***$P$ (WT vs. KO) = 0.0001. Source data are available online for this figure.

In addition, we examined the effects of Gpr54 deletion on depilation-induced synchronous hair regeneration. By day 3 post depilation, anagen was already induced in the dorsal skin of *Gpr54* KO mice (Fig. 2B). In contrast, the wild-type mice began entering into the anagen stage until day 13, by which time the KO mouse back skin had been covered with dense hairs (Fig. 2B). In order to exclude the disturbance of hormones and immunity on hair growth after *Gpr54* deletion, we transplanted *Gpr54* WT and KO skin at P19 onto the backs of nude mice (Fig. 2C). Although the hair numbers were not changed (Figs. 2D and EV2A,B), hair length in *Gpr54* KO explants was longer than in *Gpr54* WT explants (Fig. 2E).

Together, these data indicated that Gpr54 deletion accelerated anagen entry during hair cycle, promoted depilation-induced hair regrowth and facilitated transplanted hair growth.

## *Gpr54* deletion elevated the activity of DPCs in mouse hair follicles

DPCs are highly specialized mesenchymal cells that are indispensable for HF morphogenesis and postnatal hair growth cycling (Abaci et al, 2018). Alkaline phosphatase (ALP) is a commonly used marker protein for identifying DPCs and indicates hair follicle inductive capacity (Taghiabadi et al, 2020). As expected, we observed that GPR54 (green) and ALP (red) co-localized in the DP

region (Fig. 3A). In order to determine whether the accelerated hair cycle resulting from Gpr54 deletion may be associated with DPCs, we measured DPC activity by detecting the expression of markers such as ALP and Versican. The immunofluorescent results showed that the expression of ALP and Versican in primary cultured DPCs of mouse hair follicles significantly increased when *Gpr54* was knocked out (Figs. 3B,C and EV2C). Consistently, in situ results from mouse hair follicles indicated that Versican levels in the dermal papilla (DP) region were significantly higher in *GPR54* KO mice compared to WT mice (Fig. EV2D). The PCR results also demonstrated a similar trend, with the expression of ALP and Versican being significantly higher in *Gpr54* KO DPCs compared to WT cells (Fig. 3D). However, the number of DPCs and proliferative capacity did not show significant differences between *Gpr54* WT and KO mice (Fig. EV3A,B).

It is known that DPCs play its role of hair inductivity via paracrine effects that include secretion of growth factors (Seo et al, 2023). As shown in Fig. 3D, *Gpr54* knockout cells displayed increased expression of *VEGF* and *IGF-1*. Meanwhile, flow cytometry results showed that the percentage of late-stage apoptosis cells during *Gpr54* knockout DPCs was significantly lower than that in WT mice (Fig. EV3C,D). Together, these results suggested that *Gpr54* knockout significantly increased the activity of DPCs.

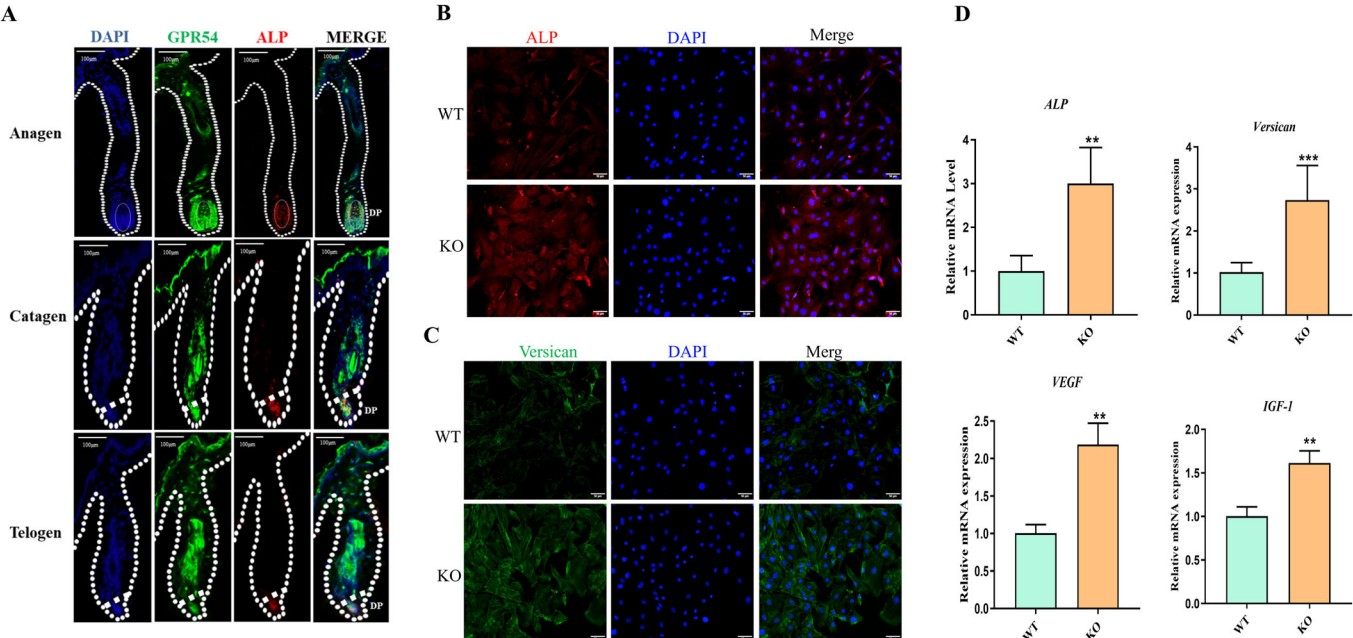

**Figure 3. Gpr54 deletion enhanced the activity of DPCs in mouse hair follicles.**

(A) Immunofluorescence labeling was performed to detect Gpr54 (green) and ALP (red) in mouse hair follicles, with nuclei counterstained using DAPI (blue). Scale bars: 100 μm. (B, C) Immunofluorescence staining of DPCs from *Gpr54* WT and KO mice labeled with antibodies against ALP (B) and Versican (C), with DAPI nuclear counterstain (blue). Scale bars: 50 μm. (D) The mRNA expression of *ALP*, *Versican*, *VEGF*, and *IGF-1* in DPCs from *Gpr54* WT and KO mice examined by Q-PCR. Data are mean ± SD of three independent experiments. A paired *t*-test was performed, $N = 3$, **$P$ (ALP) = 0.0011, *$P$ (Versican) = 0.0006, **$P$ (VEGF) = 0.0026, **$P$ (IGF-1) = 0.0038, WT versus KO. Source data are available online for this figure.

## Gpr54 deletion inhibited the expression and activity of NFATc3

NFAT proteins are calcium-regulated transcription factors that play a critical role in cell cycle, apoptosis, angiogenesis, and metastasis-related genes (Mognol et al, 2016). In mouse hair follicles, where skin stem cells reside, the NFATc1 protein regulates hair cycle and hair regrowth (Keyes et al, 2013). Therefore, we investigated whether *Gpr54* knockout affected the expression or activity of NFAT proteins. Surprisingly, in *Gpr54* knockout DPCs, the expression of NFATc3 is significantly reduced compared to WT mice, but there is no significant difference in the expression of NFATc1, NFATc2, and NFATc4 (Fig. 4A). Consistently, immunofluorescence staining showed that the expression level of NFATc3 in *Gpr54* WT mouse hair follicles is significantly higher than that in *Gpr54* KO mice. Furthermore, NFATc3 is abundantly distributed in the DP and bulge region of mouse hair follicles (Figs. 4B and EV4A).

It is reported that elevated intracellular calcium levels activate calcineurin, a calcium/calmodulin-dependent phosphatase, which dephosphorylates NFATc, leading to its nuclear translocation and the activation of target gene expression (Park et al, 2020). As expected, we observed that in *Gpr54* knockout DPC cells, the level and activity of calcineurin is significantly lower than the WT mice (Fig. 4C,D). Meanwhile, the results of nuclear-cytoplasmic fractionation showed that NFATC3 levels in the nuclei of DPCs from *Gpr54* KO mice were significantly lower than those in WT mice. In contrast, the levels of p-NFATc3 in the cytoplasm of DPC cells from *Gpr54* KO mice were higher compared to the WT mice (Fig. 4E). Cyclosporine A (CsA), a

calcineurin inhibitor, lowered the nuclear NFATc3 of *Gpr54* WT cells to levels similar to those in *Gpr54* KO cells (Fig. 4F). All the results indicated that Gpr54 knockout inhibited the expression and activity of NFATc3 in hair follicles.

## Gpr54 deletion activated Wnt signal pathway by downregulating SFRP1

Considering that NFATc3 is a gene closely relevant with the Wnt signaling pathway, we further investigated the impact of *Gpr54* gene knockout on the activity of the Wnt signaling pathway in DPCs. As shown in Fig. 5A, eight genes associated with Wnt signaling pathway including *Wnt10a*, *Wnt10b*, *Wnt3a*, *Lrp5*, *Fzd10*, *LEF1*, *Cyclin d1*, and *Axin2*, were upregulated in *Gpr54* KO DPCs compared with the normal ones. Consistently, the levels of nuclear β-catenin in *Gpr54* KO DPCs were significantly higher than that in *Gpr54* WT DPCs (Figs. 5B and EV4B). LEF1 is a downstream targeted gene of the Wnt/β-catenin signaling pathway. The results from both Western blot and immunofluorescence assays both showed a significant increase in the expression of LEF1 in *Gpr54* gene-deleted DPCs as compared to their normal counterparts (Figs. 5B,C and EV4B). Consistently, the levels of β-catenin in hair follicle cells in vivo were significantly higher in Gpr54 KO mice compared to WT mice (Fig. EV4C). These results indicated that *Gpr54* deletion activated the Wnt/β-catenin signaling pathway in DPCs.

SFRP1 was a secreted antagonist of WNT signaling pathway and could block the WNT signaling pathway. We observed a significant decrease in both the mRNA and protein levels of SFRP1 in *Gpr54*

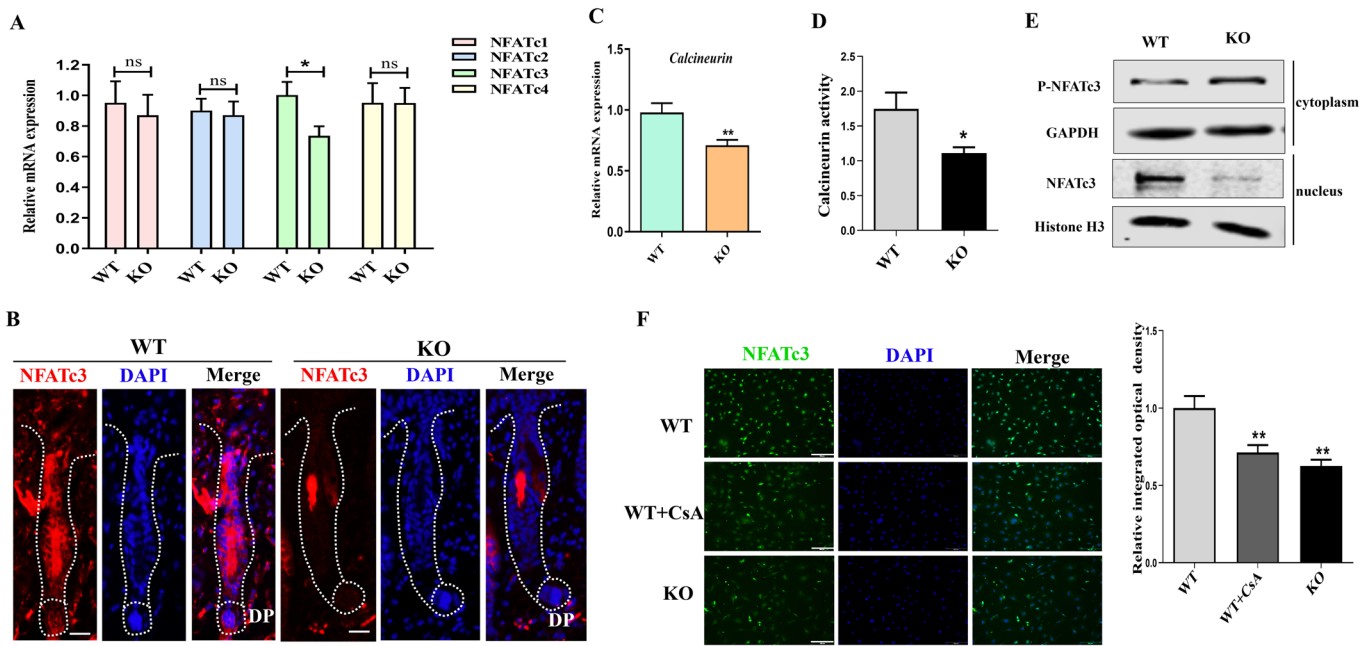

**Figure 4.** *Gpr54* deletion inhibited the expression of NFATc3 and its nuclear localization.

(A) Q-PCR analysis of *NFATc1, NFATc2, NFATc3,* and *NFATc4* in DPCs from *Gpr54* WT and KO mice. Data are mean ± SD of three independent experiments. A paired *t*-test was performed, *N* = 3, *\*P* = 0.0122, WT versus KO. (B) Immunofluorescence labeling of NFATc3 in the hair follicles of *Gpr54* WT and KO mice. DP is an abbreviation for dermal papilla. Scale bars: 50 μm. (C) The mRNA expression of *calcineurin* in DPCs from *Gpr54* WT and KO mice detected by Q-PCR. Data are mean ± SD of three independent experiments. A paired *t*-test was performed, *N* = 3, *\*\*P* = 0.0061, WT versus KO. (D) The activity of calcineurin in DPCs from *Gpr54* WT and KO mice by using a colorimetric method. Data are mean ± SD of three independent experiments. A paired *t*-test was performed, *N* = 3, *\*P* = 0.0114, WT versus KO. (E) Levels of p-NFATc3 in the cytoplasm and NFATc3 in the nucleus of *Gpr54* WT and KO DPCs as determined by nuclear-cytoplasmic separation followed by Western blotting. (F) Immunofluorescence staining of NFATc3 in *Gpr54*$^{+/+}$ DPCs treated with or without CsA (1 μM) and *Gpr54*$^{-/-}$ DPCs. The bar graph on the right represents the quantitative analysis of NFATc3 expression. Scale bars: 100 μm. Data are mean ± SD of three independent experiments. A paired *t*-test was performed, *N* = 3, *\*\*P* (WT+CsA vs. WT) = 0.0054, *\*\*P* (WT vs. KO) = 0.0018. Source data are available online for this figure.

knockout DPCs (Figs. 5A,D and EV4D). The results of the immunofluorescent experiments provided additional confirmation of the decreased levels of SFRP1 in *Gpr54* knockout DPCs (Figs. 5E and EV5A,B). Interestingly, CsA treatments significantly reduced the expression of SFRP1 in *Gpr54* WT DPCs to a level comparable to that in KO DPCs (Fig. 5D,E), suggesting that *Gpr54* deletion may activate the Wnt signaling pathway through the calcineurin-NFATc3-SFRP1 axis.

## *Gpr54* deletion activated HFSCs in mouse hair follicles

In mammals, activation of HFSCs is important for cyclical rounds of hair regeneration (Flores et al, 2017). Thus, we further detected the effects of Gpr54 deletion on HFSCs. As shown in Fig. 6A, immunofluorescent staining results showed the co-localization of Gpr54 and CD34 in the bulge region of hair follicles, suggesting that Gpr54 may influence stem cell function. Next, using FACS analysis, we observed that ITGA6$^+$/CD34$^+$ cells accounted for 10.3% of the bulge cells in *Gpr54* KO mice, whereas they constituted 7.98% of the bulge cells in WT mice (Fig. 6B). Furthermore, immunofluorescence staining reveals a significantly higher level of CD34 in *Gpr54* KO mice than in WT mice (Fig. EV5C,D). In addition, in the hair follicles of *Gpr54* KO mice, the proportion of Lgr5-positive cells in the bulges was significantly higher compared to *Gpr54* WT mice (Fig. 6C). Meanwhile, we analyzed the cell cycle of CD34$^+$ hair follicle stem cells using flow

cytometry. The results showed that the proportion of CD34$^+$ HFSCs in the S phase of the cell cycle in *Gpr54* KO mice was significantly higher than that in *Gpr54* WT mice (Fig. 6D). The data collectively demonstrated that *Gpr54* deletion not only augmented the number but also amplified the activity of mouse HFSCs.

## Gpr54 inhibitor KP234 promoted mouse hair regrowth

Given that Gpr54 knockout promotes the hair cycle process, we next investigated whether Gpr54 inhibitors would have a similar effect as gene knockout. As expected, KP234 treatment markedly elevated ALP levels in cultured DPCs in a concentration-dependent manner (Fig. 7A,B). Through in vitro cultivation of mouse hair follicles, we observed that, compared to the control group, the Gpr54 inhibitor KP234 effectively promoted the growth rate of hair shafts in mouse hair follicles (Fig. 7C). Statistical analysis also revealed that the average hair shaft length of the KP234-treated group was significantly greater than in untreated control group (Fig. 7D). After hair cycle synchronization by depilation, we observed that the fur color of mouse backs in the KP234-treated group was noticeably darker than in the untreated group, suggesting a significantly faster rate of hair regeneration in the KP234-treated group compared to controls (Fig. 7E). Thus, Gpr54 inhibitor KP234 showed promising effects in promoting hair growth or regeneration, indicating its potential for the treatments of hair loss such as androgenic alopecia in the future.

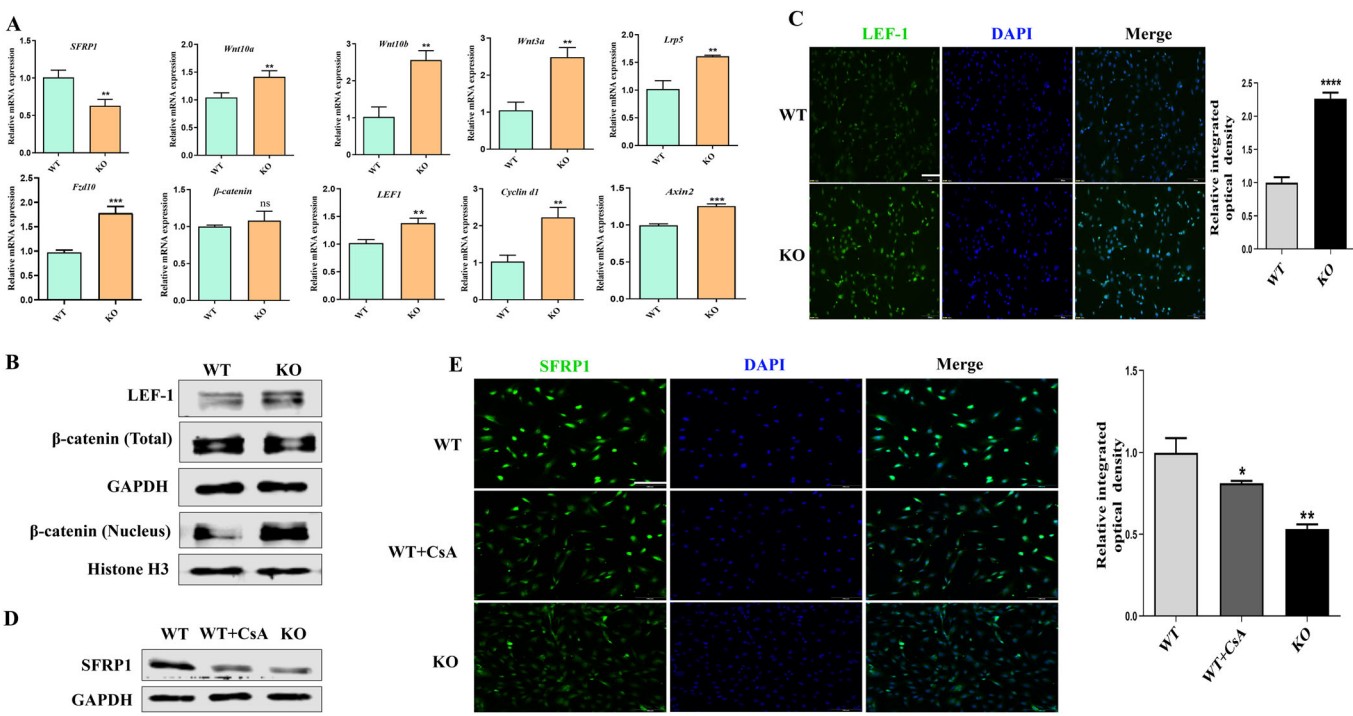

**Figure 5. Gpr54 deletion activated Wnt/β-catenin signal pathway by downregulating SFRP1.**

(A) Q-PCR analysis of SFRP1 and Wnt-related genes (Wnt10a, Wnt10b, Wnt3a, Lrp5, Fzd10, LEF1, and Axin2) in DPCs from Gpr54 WT and KO mice. Data are mean ± SD of three independent experiments. A paired t-test was performed, $N = 3$, **$P$ (SFRP1) = 0.0070, **$P$ (Wnt10a) = 0.0097, **$P$ (Wnt10b) = 0.0020, **$P$ (Wnt3a) = 0.0019, **$P$ (Lrp5) = 0.0025, ***$P$ (Fzd10) = 0.0007, **$P$ (Lef1) = 0.0049, **$P$ (Cyclin d1) = 0.0030, ***$P$(Axin2) = 0.0002, WT versus KO. (B) Immunoblot analysis of total β-catenin and LEF1 in Gpr54 WT and KO DPCs. Levels of β-catenin in the nucleus was determined by nuclear-cytoplasmic separation followed by Western blotting. (C) Immunofluorescence staining of Lef1 in DPCs from Gpr54 WT and KO mice. Scale bars: 100 μm. The bar graph on the right presents a quantitative analysis of the immunofluorescent staining of LEF1. Data are mean ± SD of three independent experiments. A paired t-test was performed, $N = 3$, ****$P < 0.0001$, WT versus KO. (D) The protein expression of SFRP1 in Gpr54 WT DPCs treated with or without CsA (1 μM) and Gpr54 KO DPCs as analyzed by Western Blot. (E) Immunofluorescence staining of SFRP1 in Gpr54 WT DPCs treated with or without CsA (1 μM) and in Gpr54 KO DPCs. The bar graph on the right represents the quantitative analysis of SFRP1 expression. Scale bars: 100 μm. Data are mean ± SD of three independent experiments. A paired t-test was performed, $N = 3$, *$P$ (WT+CsA vs. WT) = 0.0236, **$P$ (WT vs. KO) = 0.0010. Source data are available online for this figure.

## Discussion

Hair follicles (HFs), originating from the ectoderm, are a dynamic mini-organ characterized by the process of hair cycling. This cycling is clinically significant, as most hair growth disorders arise from disruptions in this chronobiological process. Although much progress has been made in understanding the molecular basis of hair loss, there remains a significant shortage of effective drugs available for the treatment of hair loss in the clinic. GPCRs are the most successful targets of modern medicine, and approximately 36% of marketed pharmaceuticals target human GPCRs (Tang et al, 2012). Currently, some specific GPCRs have been shown to be capable of activating HFSCs (Kim and Sung, 2023; Ren et al, 2020). However, the relationship between GPCRs and hair cycle has not yet been fully understood. Using a genetic mouse model, we uncovered a previously unrecognized function of Gpr54 in regulating the telogen-to-anagen transition and hair regrowth. Further analyses demonstrated that loss of Gpr54 not only increased the activity of DPCs but also promoted the proliferation of HFSCs. It achieved this function by activating Wnt/β-catenin signal pathway through the inhibition of SFRP1 expression, possibly in a calcineurin-NFATc3-dependent manner. Strikingly,

the Gpr54 inhibitor KP234 exhibited an excellent capacity in promoting hair regeneration, implying that GPR54 could be a promising drug target for the treatment of hair loss in the future.

DPCs have the potential to induce differentiation of epithelial stem cells into hairs. Although there are many studies on proteins or signaling molecules affecting the activity of HFSCs such as lymphoid enhancer factor/T-cell factor, Wnt/β-catenin, transforming growth factor-β/bone morphogenetic protein, Notch and Hedgehog (Wang et al, 2022), relatively few focus on DPCs. Our research confirms that Gpr54 is highly expressed in the DPCs of hair follicles, suggesting that Gpr54 may be a novel regulatory factor of DPCs. As the marker of the dermal papilla, ALP and Versican are highly expressed in the early anagen while their low expression was reported as the loss of the hair inductivity in dermal papilla (Taghiabadi et al, 2020). Indeed, the increase in ALP and Versican expression following GPR54 knockout confirms GPR54's inhibitory role on DPC activity. In addition, the observed enhancement of anti-apoptotic capacity and mediator secretion in DPCs upon GPR54 deletion further supports its influence on promoting hair growth. Together, we infer that the activation of DPCs in Gpr54 KO mice plays an important role in the acceleration of hair cycle and hair regeneration.

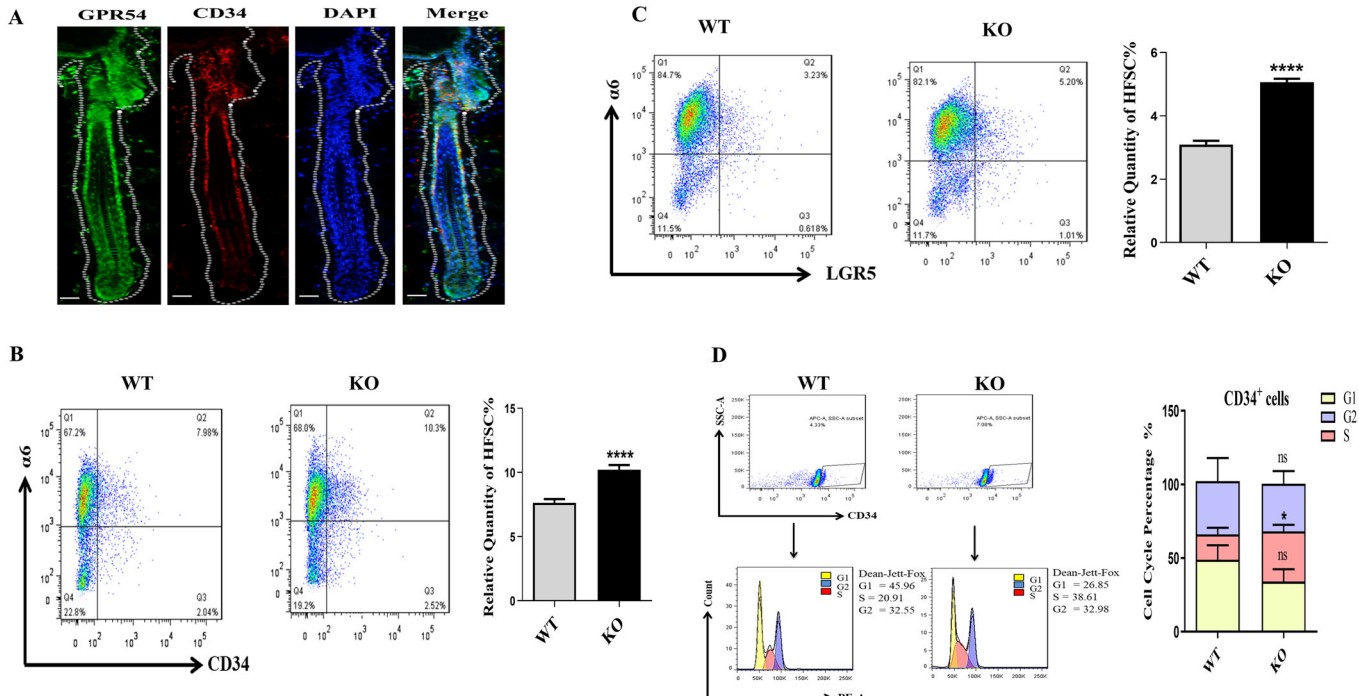

**Figure 6. *Gpr54* deletion activated HFSCs in mice.**

(A) Immunofluorescence labeling was performed to detect Gpr54 (green) and CD34 (red) in mouse hair follicles, with nuclei counterstained using DAPI (blue). Scale bars: 50 μm. (B) Flow cytometric evaluation of the quantity of CD34+ and integrin α6+ cells in the hair follicles of *Gpr54* WT and KO mice at day 19. The bar graph shows the percentage of CD34+ and integrin α6+ hair follicle stem cells. Data are mean ± SD of three independent experiments. A paired *t*-test was performed, $N = 6$, ****$P < 0.0001$, WT versus KO. (C) Flow cytometry analysis of the populations of LGR5+ and integrin α6+ cells within the hair follicles of *Gpr54* WT and KO mice. The bar chart displays the proportions of LGR5+ and integrin α6+ hair follicle stem cells. Data are mean ± SD of three independent experiments. A paired *t*-test was performed, $N = 4$, ****$P < 0.0001$, WT versus KO. (D) Cell cycle analysis of CD34+ hair follicle stem cells in the hair follicles of *Gpr54* WT and KO mice was conducted using flow cytometry. The bar graph represents the percentages of cells in each phase of the cell cycle. Data are mean ± SD of three independent experiments. A paired *t*-test was performed, $N = 3$, *$P$ (S phage) = 0.0101, WT versus KO. Source data are available online for this figure.

The Wnt signaling is the primitive dermal induction pathway for developing hair placodes and also play a central role in hair follicle regeneration (Shin, 2022). Interestingly, we find the expression of secreted frizzled related protein 1 (SFRP1), an inhibitory factor of the Wnt signaling pathway, is positively regulated by Gpr54. Meanwhile, the nuclear level of β-catenin, an effector molecule of the Wnt signaling pathway, significantly increases after *Gpr54* deletion, suggesting activation of the Wnt signaling pathway when Gpr54 is absent. LEF1, as the Wnt target, is necessary for the transduction of Wnt signaling and is crucial for the development of HF (Li et al, 2023). We also observed that Gpr54 deletion markedly upregulated the expression of Lef1 in DPCs. Thus, our results indicate that Gpr54 deletion activates the Wnt signaling pathway by regulating the expression of SFRP1 in the DPCs of hair follicles. Although kisspeptin is reported to be able to activate the pathways related to mitogen-activated protein kinases (MAPK), NFkB, and phosphatidylinositol-3-kinase (PI3K)/Akt (Castaño et al, 2009; Huang et al, 2018), its relationship with the Wnt/β-catenin pathway is still unknown. Thus, the hair follicle is the first confirmed site where Gpr54 regulates the activity of Wnt signaling pathway.

It is reported that kisspeptin/GPR54 signaling recruited calcineurin and increased its phosphatase activity in peripheral immune cells (Huang et al, 2018). Consistently, we also find that

*Gpr54* knockout decreases calcineurin expression and activity in DPCs. The classic NFAT isoforms (NFAT1–NFAT4) exist in a hyperphosphorylated state in the cytoplasm under resting conditions. They are usually activated by increased intracellular calcium levels, via dephosphorylation by the phosphatase calcineurin, which triggers the transport of NFAT proteins from the cytoplasm to the nucleus (Tong et al, 2022). Although Nfatc1 was reported to orchestrate aging in hair follicle stem cells (Keyes et al, 2013), the effect of NFATc3 on hair follicles is still unclear. Interestingly, we find that only NFATc3 can be regulated by Gpr54 in both the DP and bulge regions of hair follicles. Gpr54 deletion inhibit NFATc3 dephosphorylation and nuclear anchorage. In addition, we observed that the NFATc3 immunostaining signal is almost absent in the hair follicles of *Gpr54* KO mice. We hypothesize that phosphorylated NFATc3 (p-NFATc3) may decrease its stability (Hanaki et al, 2024), leading to reduced NFATc3 staining. However, WB results in Fig. 4E showed higher p-NFATc3 levels in DPCs of KO mice compared to WT mice, which we speculate may be due to Western blot capturing phosphorylation at a specific time point, allowing detection of the phosphorylated protein before complete degradation. Using calcineurin inhibitor CsA, we indeed observed that calcineurin targets and dephosphorylates NFATc3. Thus, we propose the possible existence of a regulatory mechanism by which the kisspeptin/GPR54/calcineurin axis modulates

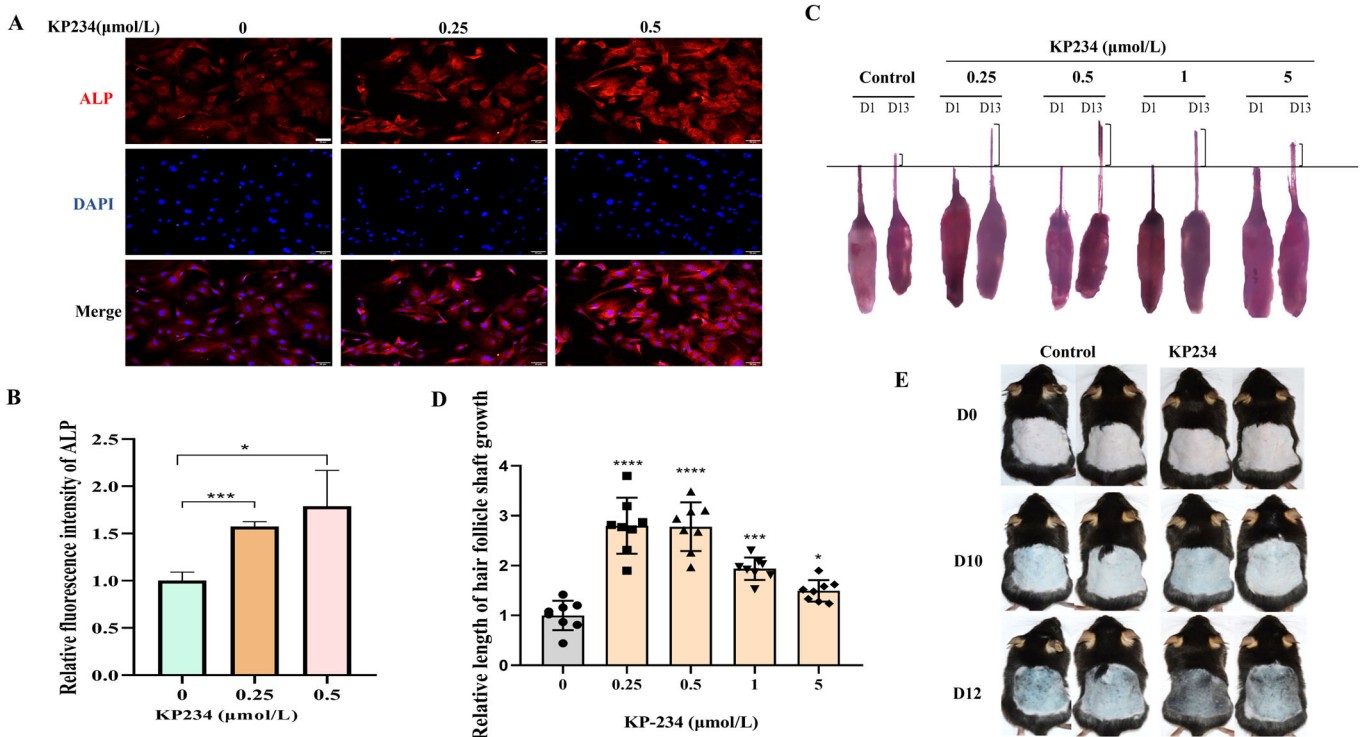

**Figure 7. KP-234 promoted hair growth and inhibited DHT-induced hair Loss.**

(A) Immunofluorescence staining of DPCs, either untreated or treated with 0.25 or 0.5 μmol/L KP-234 for 24 h, labeled with antibodies against ALP (red) and counterstained with DAPI for nuclei (blue). Scale bars: 50 μm. (B) Quantitative analysis of ALP fluorescence intensity for Fig. 7A. Data are expressed as mean ± SD of three independent experiments. A paired $t$-test was performed, $N = 3$, ***$P$ (0.25 μM vs. 0 μM) = 0.0007, *$P$ (0.5 μM vs. 0 μM) = 0.0246. (C) The representative photos of cultured mouse vibrissae hair follicles at day 1 and day 13. The isolated vibrissae hair follicles were incubated with KP-234 at concentrations ranging from 0.25 to 5 μmol/L or with a control vehicle for 13 days. (D) The length of hair shaft growth in mouse vibrissae hair follicles cultured with KP-234 at concentrations ranging from 0.25 to 5 μmol/L, or with a control vehicle, measured on day 13. The data are expressed as mean ± SD. A paired $t$-test was performed, $N = 8$. ****$P$ (0.25 μM vs. 0 μM) < 0.0001, ****$P$ (0.5 μM vs. 0 μM) < 0.0001, ***$P$ (1 μM vs. 0 μM) = 0.0002, *$P$ (5 μM vs. 0 μM) = 0.0229. (E) KP-234 promoted synchronized hair regrowth in mice induced by depilation. C57BL/6 mice, after depilation, were treated daily with a 20 μmol/L concentration of KP-234. Source data are available online for this figure.

NFATc3 activity within hair follicles. It is known that NFATc3 proteins can collaborate with other factors to activate transcription of its downstream genes. We thus assumed that SFRP1, a Wnt signaling pathway antagonist, maybe a new target gene of NFATc3, which need further investigation in the future. Microarray analysis identified SFRP1 as being downregulated in the dermal papilla (DP) of CsA-treated human scalp hair follicles (HFs) ex vivo (Hawkshaw et al, 2018), which further provides evidence for our conclusions.

HFSCs serve as a reservoir of progenitor cells that are essential for the regeneration of hair follicles. During the transition from telogen to anagen, HFSCs become activated, proliferate, and migrate to replenish the epidermal components of hair follicles, such as hair matrix cells (Sun et al, 2024). Interestingly, we observed the increase in the ratios of HFSCs in the S phase and the number of Lgr5+ active stem cells following Gpr54 deletion, indicating that Gpr54 is also involved in the regulation of stem cell proliferation and differentiation. Importantly, in the bulge region of Gpr54 knockout (KO) mouse hair follicles, the levels of NFATc3 are significantly lower, while β-catenin levels are significantly higher compared to wild-type (WT) mice. Thus, we assume that deletion of Gpr54 may also activate HFSCs by NFATC3-SFRP1-Wnt-β-catenin signaling cascades, which need further investigation in the future. Meanwhile, we observed that some mediators such as

Wnts (Wnt3a, Wnt10a, Wnt10b), IGF-1 and VEGF were significantly upregulated in DPCs when Gpr54 was knockout. These factors, secreted by DPCs into the stem cell niche, can activate HFSCs, further supporting the role of GPR54 deletion in promoting HFSC activation. In Gpr54 KO mice, this activation ultimately accelerated the entry of hair follicles into a new anagen phase. A previous study has shown that HFSCs may experience increased DNA damage and aging risk during frequent hair cycle turnover, potentially affecting the long-term stability of the stem cell pool (Matsumura et al, 2016). At the same time, other research indicates that HFSCs possess a high capacity for self-renewal and can maintain their quantity and function in frequent hair cycles by adjusting asymmetric division or self-renewal strategies, thereby preventing stem cell pool depletion (Matsumura et al, 2021). Whether Gpr54 knockout, which accelerates the normal hair cycle process, impacts the stability of the HFSC pool requires further investigation.

Together, we identify a novel function of GPR54 in regulating hair cycle and promoting hair regeneration. Our study also demonstrates that Gpr54 can activate DPCs and HFSCs by regulating the NFATC3-SFRP1-Wnt-β-catenin signaling pathway (Fig. 8). Critically, KP234, a specific inhibitor for Gpr54, can effectively speed up hair shaft growth in vitro and promote hair regeneration in vivo. Thus, our study

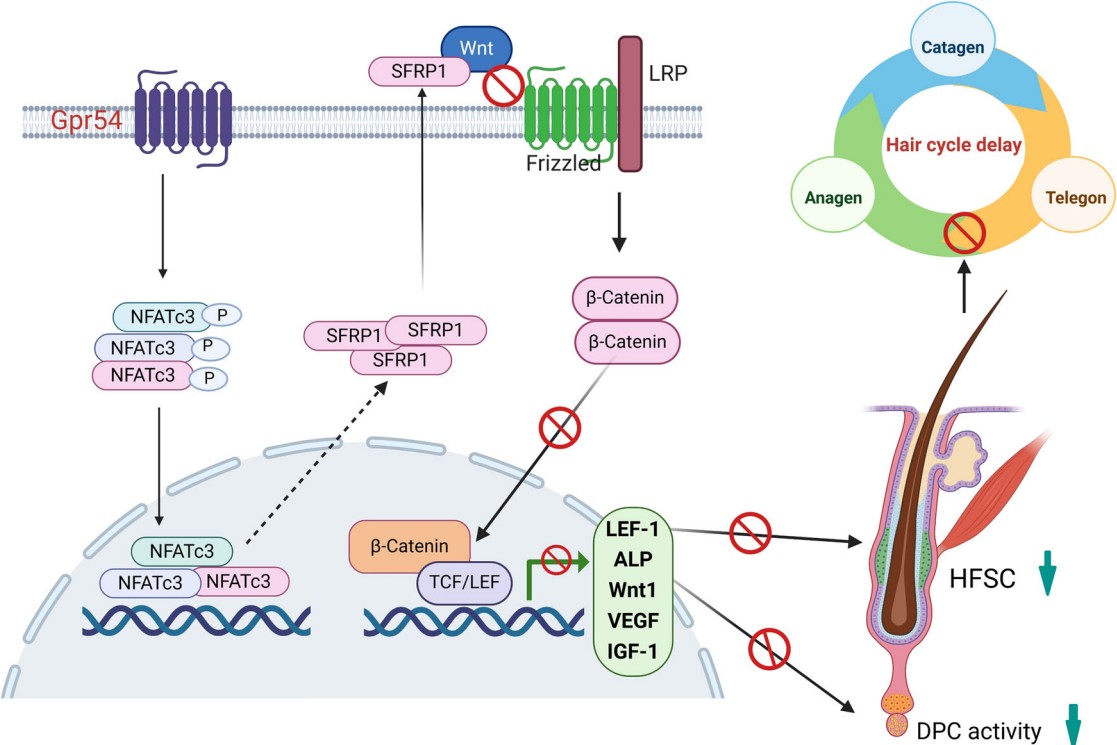

**Figure 8. Schematic diagram of Gpr54 regulation in hair cycle and hair regeneration.**

Gpr54 activates calcineurin, leading to NFATc3 dephosphorylation and increased SFRP1 expression. SFRP1 disrupts the Wnt ligand-receptor interaction, reducing β-catenin activity and altering the activity of DPCs and HFSCs, which in turn influences hair cycle and regeneration.

potentially provides a new avenue for developing drugs that target Gpr54 to treat hair disorders in the future.

# Methods

### Reagents and tools table

| Reagent/ Resource | Reference or Source | Identifier or Catalog Number |
|---|---|---|
| **Experimental models** | | |
| C57BL/6 (*M. musculus*) | Experimental Animal Center of East China Normal University | N/A |
| *GPR54*⁻/⁻ (*M. musculus*) | Experimental Animal Center of East China Normal University | N/A |
| **Recombinant DNA** | | |
| **Antibodies** | | |
| Anti-GPR54 | Affinity | DF7123 |
| Anti-ALP | Abcam | 65834 |
| Anti-Versican | Abcam | 19345 |
| Anti-NFATc3 | Santa Cruz Biotechnology | 7294 |
| Anti-LEF1 | Santa Cruz Biotechnology | 374522 |
| Anti-SFRP1 | Proteintech | 26460-1-AP |
| Anti-CD34 | Proteintech | 11026-1-AP |

| Reagent/ Resource | Reference or Source | Identifier or Catalog Number |
|---|---|---|
| Anti-Histone H3 | CST | 9715 |
| Anti-GAPDH | Proteintech | 60004-1-Ig |
| **Oligonucleotides and other sequence-based reagents** | | |
| Mus.*GPR54* | This study | F: 5′-TACATCGCTA ACCTGGCTGC-3′ |
| Mus.*GPR54* | This study | R: 5′-CCCAGATGCT GAGGCTGAC-3′ |
| Mus.*Kiss1* | This study | F: 5′-CCGTCCAACGC TTCAGGAT-3′ |
| Mus.*Kiss1* | This study | R: 5′-GTGTAGCGA AAAACAGGGGAA-3′ |
| Mus.*NFATc1* | This study | F: 5′-GGAGAGTCCG AGAATCGAGAT-3′ |
| Mus.*NFATc1* | This study | R: 5′-TTGCAGCTAGG AAGTACGTCT-3′ |
| Mus.*NFATc2* | This study | F: 5′-CTCGGCCTTTG CCCATCTC-3′ |
| Mus.*NFATc2* | This study | R: 5′-AGGAGCACGG AGCATCTGA-3′ |
| Mus.*NFATc3* | This study | F: 5′-GCTCGACTTCA AACTCGTCTT-3′ |
| Mus.*NFATc3* | This study | R: 5′-GATGTGGTAAG CCAAGGGATG-3′ |

| Reagent/ Resource | Reference or Source | Identifier or Catalog Number |
|---|---|---|
| Mus.*NFATc4* | This study | F: 5′-GAGCTGGAATT TAAGCTGGTGT-3′ |
| Mus.*NFATc4* | This study | R: 5′-GGAGGGGTAT CCTCTGAGTCC-3′ |
| Mus. *Calcineurin* | This study | F: 5′-AGGCTCATTCCAG ACACCG-3′ |
| Mus. *Calcineurin* | This study | R: 5′-GTCCTGCTCAGGTT TAGCCG-3′ |
| Mus. *ALP* | This study | F: 5′-CCAACTCTTTT GTGCCAGAGA-3′ |
| Mus. *ALP* | This study | R: 5′-GGCTACATTGG TGTTGAGCTTTT-3′ |
| Mus. *Versican* | This study | F: 5′-TTTTACCCGAGT TACCAGACTCA-3′ |
| Mus. *Versican* | This study | R: 5′-GGAGT AGTTGT TACATCCGTTGC-3′ |
| Mus.*Wnt3a* | This study | F: 5′-CTCCTCTCGGA TACCTCTTAGTG-3′ |
| Mus.*Wnt3a* | This study | R: 5′-CCAAGGACCA CCAGATCGG-3′ |
| Mus.*Wnt10a* | This study | F: 5′-CATGCTCGAAT GAGACTCCAC-3′ |
| Mus.*Wnt10a* | This study | R: 5′-CCCTACTGTGC GGAACTCAG-3′ |
| Mus.*Wnt10b* | This study | F: 5′-ATCGCCGTTCAC GAGTGTC-3′ |
| Mus.*Wnt10b* | This study | R: 5′-GGAAACCGCGCT TGAGGAT-3′ |
| Mus.*Lrp5* | This study | F: 5′-ACGTCCCGTAA GGTTCTCTTC-3′ |
| Mus.*Lrp5* | This study | R: 5′-GCCAGTAAATGT CGGAGTCTAC-3′ |
| Mus.*Fzd10* | This study | F: 5′-CATGCCCAAC CTGATGGGTC-3′ |
| Mus.*Fzd10* | This study | R: 5′-GCCACCTGAAT TTGAACTGCTC-3′ |
| Mus.*β-catenin* | This study | F: 5′-GAAGAGATGCC GGTTTGTTGA-3′ |
| Mus.*β-catenin* | This study | R: 5′-GCCCGAAGCT CCATCACTC-3′ |
| Mus.*Lef1* | This study | F: 5′-GCCACCGATGAG ATGATCCC-3′ |
| Mus.*Lef1* | This study | R: 5′-TTGATGTCGGC TAAGTCGCC-3′ |
| Mus.*CyclinD1* | This study | F: 5′-GCGTACCCTGA CACCAATCTC-3′ |
| Mus.*CyclinD1* | This study | R: 5′-ACTTGAAGTA AGATACGGAGGGC-3′ |
| Mus.*Axin2* | This study | F: 5′-AACCTATGCCCGT TTCCTCTA-3′ |
| Mus.*Axin2* | This study | R: 5′-GAGTGTAAAGA CTTGGTCCACC-3′ |
| Mus.*sFRP1* | This study | F: 5′-TCTAAGCCCCAAG GTACAACC-3′ |
| Mus.*sFRP1* | This study | R: 5′-GCTTGCACAGAG ATGTTCAATG-3′ |
| Mus. *VEGF* | This study | F: 5′-CTGCCGTCCGAT TGAGACC-3′ |
| Mus. *VEGF* | This study | R: 5′-CCCCTCCTTGTA CCACTGTC-3′ |

| Reagent/ Resource | Reference or Source | Identifier or Catalog Number |
|---|---|---|
| Mus. *IGF1* | This study | F: 5′-CTGGACCAGAGA CCCTTTGC-3′ |
| Mus. *IGF1* | This study | R: 5′-GGACGGGGACT TCTGAGTCTT-3′ |
| **Chemicals, Enzymes and other reagents** | | |
| William's Medium E | Gibco | 12551032 |
| Collagenase I | Gibco | 17100017 |
| Fetal Bovine Serum | Gibco | A5669701 |
| Penicillin-streptomycin solution | Gibco | 15140122 |
| BCA assay | TaKaRa | T9300A |
| Magzol Reagent | Magen | KD210300 |
| PrimeScript™ RT reagent Kit | TaKaRa | RR037Q |
| qPCR SYBR Green Master Mix | Yeasen | 11201ES03 |
| MTS | Promega | G5421 |
| Crystal violet staining solution | Beyotime | C0121 |
| Calcineurin assay kits | Enzo | BML-AK804 |
| Kisspeptin 234 | Tocris | 3881/1 |
| **Software** | | |
| Graph Pad Prism v8.4.3 | https://www.graphpad.com/ | |
| ImageJ/Fiji | https://imagej.net/software/fiji/ | |
| Photoshop | https://www.adobe.com/pt/products/photoshop.html | |
| **Other** | | |

## Animals

The 8-week-old *GPR54*[+/+] and *GPR54*[−/−] mice were obtained from the SPF-grade Experimental Animal Center of East China Normal University, and were provided with ad libitum access to food and water. *Gpr54*-deficient mice (C57BL/6) were generated as previously described(Funes et al, 2003). Primers used for the identification of mutated mice are 5′-GCCTAAGTTTCTCTGGTGGAGGATG-3′ (wild-type), 5′-GTGGGATTAGATAAATGCCTGCTCT-3′ (knockout), and 5′-CGCGTACCTGCTGGATGTAGTTGAC-3′ (common). All the animal experimental protocols were approved by the Ethics Committee of East China Normal University.

## Isolation and culture of mouse hair follicles and primary DPCs

After anesthetizing 8-week-old C57BL/6 mice, the skin of their whiskers was taken, and hair follicles were isolated using tweezers. The separated mouse vibrissae follicles were then cultured in

William's Medium E (Gibco, 12551032, New York, NY, USA) and incubated with Kisspeptin-234 at concentrations ranging from 0.25 to 5 μmol/L or a control vehicle for 13 days.

For the isolation of DPCs, the hair bulbs of $GPR54^{+/+}$ and $GPR54^{-/-}$ mice were cut from the lower end of the obtained follicles. They were digested for 1 h with 0.2% Collagenase I (Gibco, 17100017, New York, NY, USA) and filtered through a 40 μm mesh to remove cells. The remaining tissue was transferred to DMEM containing 15% Fetal Bovine Serum (FBS) (Gibco, New York, NY, USA) and 1% penicillin-streptomycin solution (Gibco, New York, NY, USA) for cultivation until a substantial number of DPCs were obtained.

## Hematoxylin-Eosin (H&E) staining

Mouse skin tissues were fixed in 4% paraformaldehyde for 24 h, rinsed with running water, and then dehydrated in a graded series of ethanol. The tissues were then cleared in xylene, embedded in paraffin, and sectioned into 5-μm-thick slides. The paraffin sections were deparaffinized in xylene, rehydrated through a graded ethanol series, and stained with hematoxylin and eosin (Beyotime). The sections were then observed and photographed under an Olympus microscope (Olympus, Tokyo, Japan).

## Immunofluorescence staining

DPCs were seeded in a 24-well plate at a density of $3 \times 10^4$. The cells were then washed three times with PBS, fixed with 4% paraformaldehyde solution for 15 min, permeabilized with 0.1% Triton X-100 for 10 min, and blocked with 1% BSA for 30 min. Primary antibodies were incubated overnight at 4 °C, followed by a 2-h incubation with fluorescent secondary antibodies and a 5-min DAPI staining. Images were taken using an Olympus microscope.

For tissue immunofluorescence, the sections were first deparaffinized, rehydrated, and antigen-retrieved with 1×EDTA at 100 °C for 20 min, followed by a 30-min blocking with 1% BSA at room temperature. The subsequent steps were the same as for cellular immunofluorescence. The primary antibodies were as follows: anti-GPR54 antibody (1:200, Affinity, DF7123, Cambridge, UK), anti-ALP antibody (1:100, Abcam, 65834, Cambridge, UK), anti-Versican antibody (1:100, Abcam, 19345, Cambridge, UK), anti-NFATc3 antibody (1:100, Santa Cruz Biotechnology, 7294, Santa Cruz, CA, USA), anti-LEF-1 antibody (1:100, Santa Cruz Biotechnology, 374522, Santa Cruz, CA, USA), anti-SFRP1 antibody (1:100, Proteintech, 26460-1-AP, Chicago, IL, USA), and anti-CD34 antibody (1:100, Proteintech, 11026-1-AP, Chicago, IL, USA).

## Western blotting analysis

To collect cellular proteins, 100 μL of RIPA lysis buffer (25 mM Tris pH 7.6, 150 mM NaCl, 1% NP-40, 0.1% SDS, 1.0% Triton X-100, 1% deoxycholate, 5 mM EDTA with protease/phosphatase inhibitors) was added to the cell samples. The protein concentration was then quantified using a BCA assay (TaKaRa, T9300A, Shanghai, China). Proteins were separated by SDS-PAGE and transferred to a nitrocellulose membrane (NC membrane). After blocking with 5% skim milk for 1 h, the membrane was incubated overnight with primary antibodies at 4 °C on a shaker, followed by a 10-min rewarming, and then a 1-h incubation with fluorescently labeled secondary antibodies. Visualization was performed

using the LI-COR Odyssey Infrared Imaging System (LI-COR, Lincoln, NE, USA). The primary antibodies used in the Western blot were as follows: anti-SFRP1 antibody (1:1000, Proteintech, 26460-1-AP, Chicago, IL, USA), anti-β-catenin antibody (1:1000, Proteintech, 51067-2-AP, Chicago, IL, USA), anti-P-NFATc3 antibody (1:1000, Santa Cruz Biotechnology, 8405, Santa Cruz, CA, USA), anti-LEF-1 antibody (1:1000, Santa Cruz Biotechnology, 374522, Santa Cruz, CA, USA), anti-NFATc3 antibody (1:1000, Santa Cruz Biotechnology, 7294, Santa Cruz, CA, USA), anti-Histone H3 antibody (1:1000, Cell Signaling Technology, 9715, Danvers, MA, USA), and anti-GAPDH antibody (1:10,000, Proteintech, 60004-1-Ig, Chicago, IL, USA).

For the separation of cytoplasmic and nuclear fractions, a nuclear and cytoplasmic extraction kit (Beyotime) supplemented with a mixture of protease and phosphatase inhibitors (Thermo Scientific, USA) was used. GAPDH was employed as a loading control for both whole-cell lysate and cytoplasmic fractions, while histone 3 was used as the loading control for the nuclear fraction.

## Quantitative real-time PCR

Total RNA of DPCs or skin tissues was isolated using Magzol Reagent (Magen, KD210300, Guangzhou, China). Reverse transcription was carried out using the PrimeScript™ RT reagent Kit (TaKaRa-Clontech, Kusatsu, Japan). Q-PCR was performed by using SYBR Green Master Mix (Yeasen, Shanghai, China) at 95 °C for 3 min followed by 40 cycles at 95 °C for 15 s, 60 °C for 30 s, and then 72 °C for 30 s.

## MTS assay

DPCs were seeded in a 96-well plate at 5000 cells per well. When cells reached 80% confluence, the medium was replaced with 120 μL of a pre-mixed culture medium, containing 100 μL of fresh medium and 20 μL of MTS (Promega, G5421, Madison, WI, USA). After incubating at 37 °C for 1 h, the absorbance was measured at 490 nm using a SPECTRA MAX 190 spectrophotometer (Molecular Devices, San Jose, CA, USA).

## Colony formation assay

DPCs were digested with 0.25% trypsin, and seeded in a 6-well plate at densities of 1000 or 2000 cells per well. The cells were continuously cultured for two weeks, with the medium being changed every 2 days. Once visible colonies appeared in the culture dishes, they were fixed with 4% paraformaldehyde for 15 min. The colonies were then stained for 20 min at room temperature using 1 mL of crystal violet staining solution (Beyotime, C0121, Shanghai, China). After washing and drying, colonies were photographed and counted using a digital camera (Nikon, Tokyo, Japan).

## Flow cytometric sorting of hair follicle stem cells

After euthanasia, mouse skin samples were collected and treated with a dissociation enzyme for overnight digestion at 4 °C. The next day, the epidermis and dermis were separated. The epidermal tissue was minced and digested for 15 min, then filtered to collect cells. The cells were incubated with antibodies on ice in the dark for 1 h, with shaking every 10 min for mixing. After staining, cells were washed twice with HBSS, the supernatant was discarded, and the cells were resuspended into sterile flow tubes pre-incubated overnight with culture medium for flow cytometric sorting.

## Flow cytometry analysis

To analyze hair follicle stem cell quantities, epidermal cells were collected. They were then incubated for 30 min at 37 °C in a staining buffer containing CD34, α6, and LGR5. The cells were washed twice with HBSS, the supernatant was discarded, and they were resuspended. The quantity of hair follicle stem cells was analyzed using a flow cytometer (BD FACS Aria II, USA).

For cell cycle analysis, $1 \times 10^6$ cells were fixed in 70% precooled ethanol at $-20\,°C$ overnight. The cells were then incubated for 30 min at 37 °C in a staining buffer containing 0.1% Triton X-100, 0.2 mg/mL RNase A, and 0.02 mg/mL PI. Subsequently, the cells were analyzed using a flow cytometer.

For apoptosis analysis, $1 \times 10^6$ cells were washed three times in precooled PBS and then resuspended in 100 μL of 1× binding buffer. Each tube was then added with 5 μL of Annexin V-Alexa Flour 647 and 10 μL of PI, followed by mixing and incubating for 15 min in the dark. Subsequently, the cells were analyzed using a flow cytometer.

## Calcineurin activity

Calcineurin activity was measured using calcineurin assay kits. Cell lysates were prepared and centrifuged at $12,000 \times g$ for 15 min at 4 °C to obtain the supernatant. Protein concentration was determined using the Bradford assay. Equal protein amounts were incubated with RII phosphopeptide substrate in a reaction buffer (50 mM Tris-HCl, pH 7.5, 100 mM NaCl, 0.5 mM DTT, 1 mM $CaCl_2$, 1 mM $MgCl_2$) at 30 °C for 30 min. The reaction was stopped with 10 mM EDTA, and phosphate release was quantified by adding malachite green reagent and measuring absorbance at 620 nm. Calcineurin activity was expressed as nmol of phosphate released per minute per mg of protein.

## Construction of synchronized hair regrowth and KP234 treatments

For the model of synchronized hair regrowth, 7-week-old wild-type (WT) C57BL6 mice were depilated on the backs. Then, they received daily subcutaneous injections of either PBS (negative control) or 20 μM of KP234 (drug group), each at 100 μL. The hair growth of the mice is then observed daily and photographed at 10 and 12 days.

## Transplantation assay

BALB/C recipient nude mice at 8-week-old and donor C57BL/6 mice at P19 were anesthetized with 0.5% pentobarbital sodium at 30 mg/kg. Their dorsal hair of C57BL/6 mice was shaved, and 8 mm punch biopsies were made at corresponding sites. The excised donor skin was placed in saline, and the dermal connective tissue was removed and washed three times. The donor skin was then transplanted onto the recipient mice and secured with six sutures. The wound area was moistened with 0.1% penicillin and wrapped in gauze. Sutures were removed 7–10 days later, and the transplanted hair growth was observed.

## Statistical analysis

Experimental data were statistically analyzed using GraphPad Prism software. The quantitative analyses of the immunofluorescence and WB experiment results were conducted in a blind manner. The results are presented as mean ± standard deviation (mean ± SD). Differences between groups were compared using a two-tailed paired $t$-test. A $p$-value indicates the significance of the differences, where $*p < 0.05$, $**p < 0.01$, $***p < 0.001$, and ns indicates no statistical difference ($p \geq 0.05$).

## Data availability

We have not generated data in this manuscript that could be deposited in public databases.

The source data of this paper are collected in the following database record: biostudies:S-SCDT-10_1038-S44319-024-00327-y.

## Peer review information

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

## Acknowledgements

This work was supported by the National Natural Science Foundation of China (82273072 and 81972561).

## Author contributions

**Weili Xia**: Data curation; Investigation; Methodology. **Caibing Wang**: Data curation; Methodology. **Biao Guo**: Data curation; Investigation. **Zexin Tang**: Software; Investigation. **Xiyun Ye**: Formal analysis. **Yongyan Dang**: Conceptualization; Resources; Supervision; Funding acquisition; Writing—review and editing.

Source data underlying figure panels in this paper may have individual authorship assigned. Where available, figure panel/source data authorship is listed in the following database record: biostudies:S-SCDT-10_1038-S44319-024-00327-y.

## Disclosure and competing interests statement

The authors declare no competing interests.

# Expanded View Figures

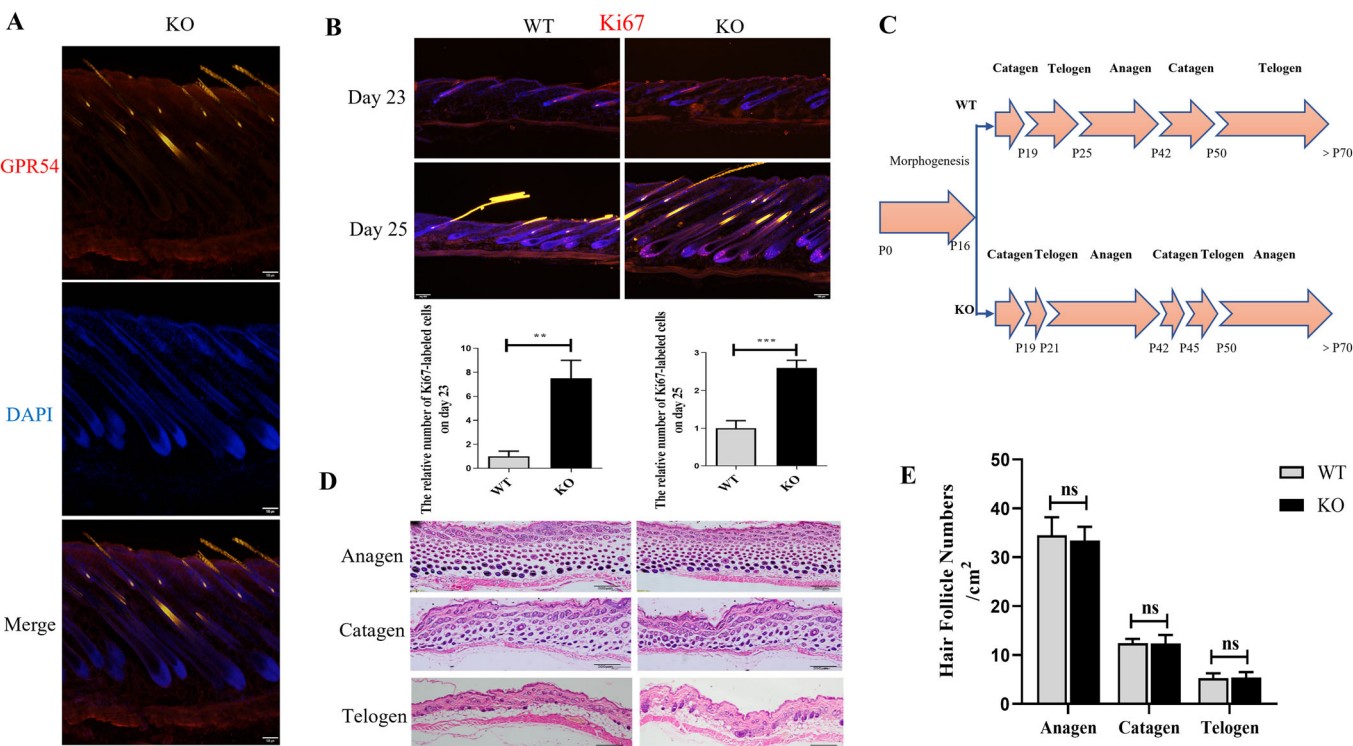

**Figure EV1.** *Gpr54* knockout accelerated the transition from telogen to anagen of mouse hair follicles.

(A) GPR54 expression was absent in *Gpr54* KO mouse hair follicles. The representative images of immunofluorescent staining for GPR54. Scale bar = 100 μm. (B) Cell proliferation in *Gpr54* WT and KO mouse hair follicles at day 23 and 25 after birth by using Ki67 immunofluorescent staining. Scale bar = 100 μm. Below the fluorescence images, the bar graphs show the quantitative analysis of the number of Ki67-positive cells in hair follicles of *Gpr54* WT and KO mice on day 23 and 25, respectively. Data are represented as mean ± SD of three independent experiments. A paired *t*-test was performed, *N* = 3. **P (P23) = 0.0020, ***P (P25) = 0.0006, WT versus KO. (C) The flowchart dynamically displays the changes in the anagen, catagen, and telogen phases of the hair cycle in *Gpr54* WT and KO mice. (D) *Gpr54* knockout did not affect the number of hair follicles in mice. HE staining of hair follicle morphology in anagen, catagen, and telogen phases in *Gpr54* WT and KO mice. (E) A quantitative analysis for Fig. EV1D, comparing the number of hair follicles in the anagen phase between *Gpr54* WT and KO mice. The data are representative of three independent experiments (mean ± SD). A paired *t*-test was performed, *N* = 3. Ns indicates no significant difference. Source data are available online for this figure.

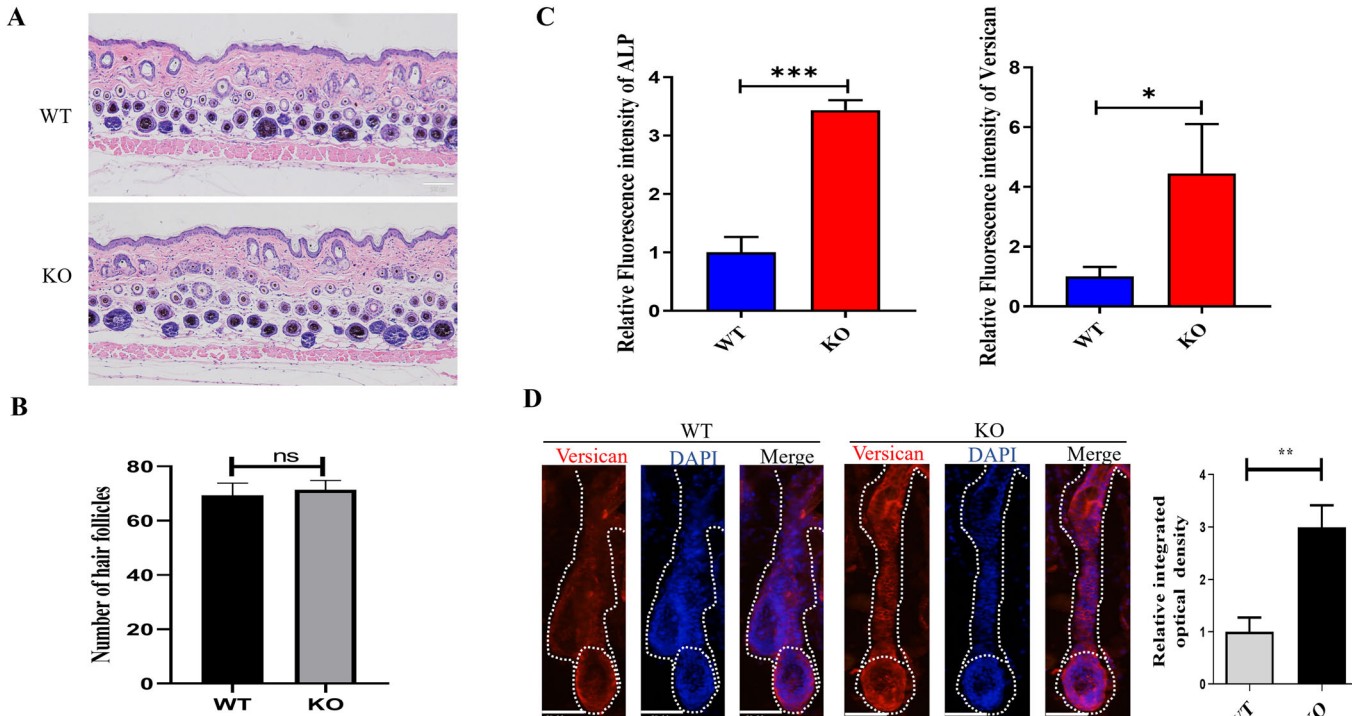

**Figure EV2. *Gpr54* knockout enhanced the activity of DPCs.**

(**A**) *Gpr54* knockout did not affect the number of hair follicles in transplants placed in BABL/C nude mice. HE staining of hair follicle morphology in transplants from *Gpr54* WT and KO mice. Scale bar = 100 μm. (**B**) A quantitative analysis for Fig. EV2A, comparing the number of hair follicles in transplants between *Gpr54* WT and KO mice. Data are expressed as mean ± SD of three independent experiments. A paired *t*-test was performed, *N* = 3. Ns indicates no significant difference. (**C**) The quantitative analysis for fluorescent intensity of ALP and Versican in Fig. 3B, C. Data are mean ± SD of three independent experiments. A paired *t*-test was performed, *N* = 3. ***P*(ALP) = 0.0002, *P*(Versican) = 0.0240, WT versus KO. (**D**) Immunofluorescence labeling of Versican in the hair follicles of *Gpr54* WT and KO mice at day 25 after birth. Scale bar = 50 μm. The bar chart on the right presents a quantitative analysis of the immunofluorescent staining of Versican. Data are mean ± SD of three independent experiments. A paired *t*-test was performed, with *N* = 3. **P* = 0.0024, WT versus KO. Source data are available online for this figure.

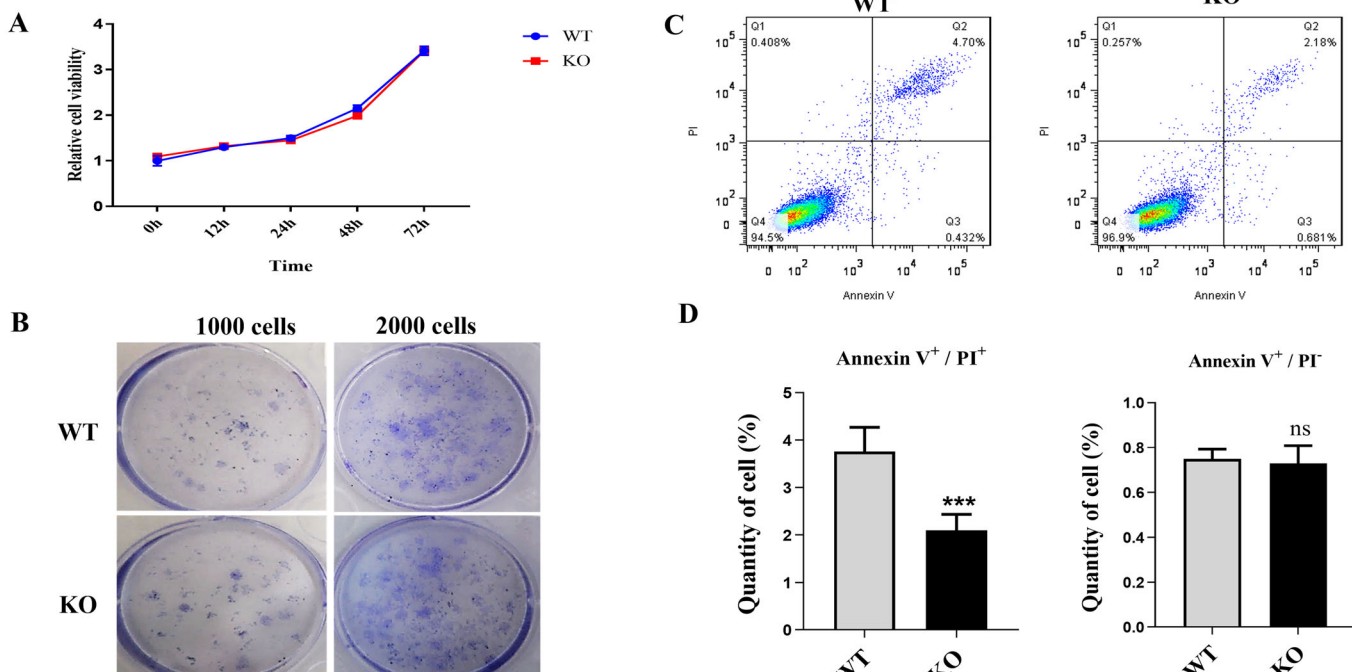

**Figure EV3. *Gpr54* knockout unchanged DPC cell number but reduced their late-stage apoptosis.**

(A) Cell proliferation in DPCs of *Gpr54* WT and KO mice assessed using the MTS assay. (B) The representative images of single-cell clone proliferation, stained with crystal violet. (C) Flow cytometry analysis with Annexin V-PI staining was performed to evaluate the percentage of apoptotic cells in DPCs of *Gpr54* WT and KO mice. (D) Quantitative analysis of the proportion of Annexin $V^+$ and $PI^+$ apoptotic cells among DPCs of *Gpr54* WT and KO mice. Data are representative of three independent experiments (mean ± SD). A paired *t*-test was performed, $N = 5$. ***$P = 0.0003$, WT versus KO. Source data are available online for this figure.

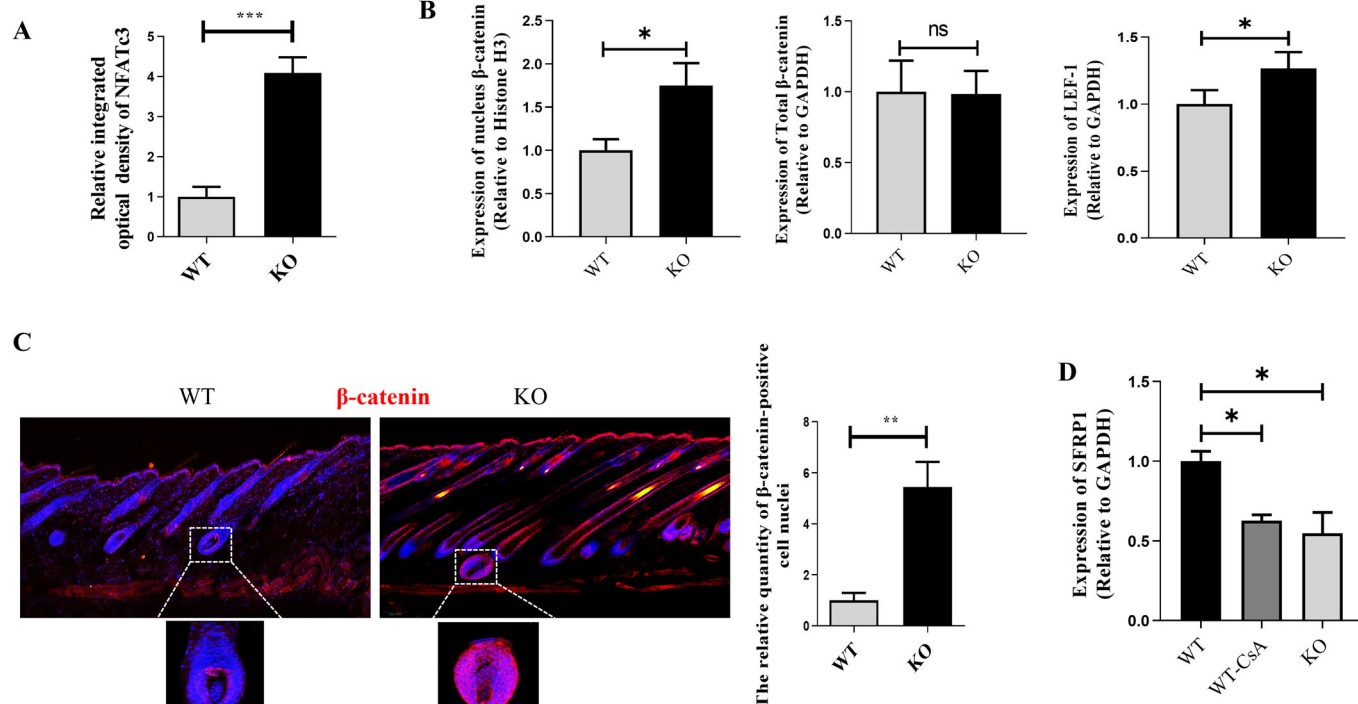

**Figure EV4. *Gpr54* deletion increased the activity of β-catenin in hair follicle cells.**

(A) A quantitative analysis for the immunofluorescent staining of NFATc3 in Fig. 4B. Data are mean ± SD of three independent experiments. A paired *t*-test was performed, $N = 3$. ***$P = 0.0003$, WT versus KO. (B) A quantitative analysis for the WB results of Fig. 5B. Data are mean ± SD of three independent experiments. A paired *t*-test was performed, $N = 3$. *$P$(β-catenin) = 0.0108, *$P$(LEF1) = 0.0444, WT versus KO. Ns indicates no significant difference. (C) Immunofluorescence staining of β-catenin in the hair follicles of *Gpr54* WT and KO mice at day 25 after birth. Scale bar = 100 µm. The bar graph on the right shows the quantitative analysis of the number of β-catenin-positive cells in the nucleus. Data are mean ± SD of three independent experiments. A paired *t*-test was performed, $N = 3$. **$P = 0.0017$, WT versus KO. (D) A quantitative analysis for the WB results of Fig. 5D. Data are mean ± SD of three independent experiments. A paired *t*-test was performed, $N = 3$, *$P$ (WT+CsA vs. WT) = 0.0179, *$P$ (WT vs. KO) = 0.0474. Source data are available online for this figure.

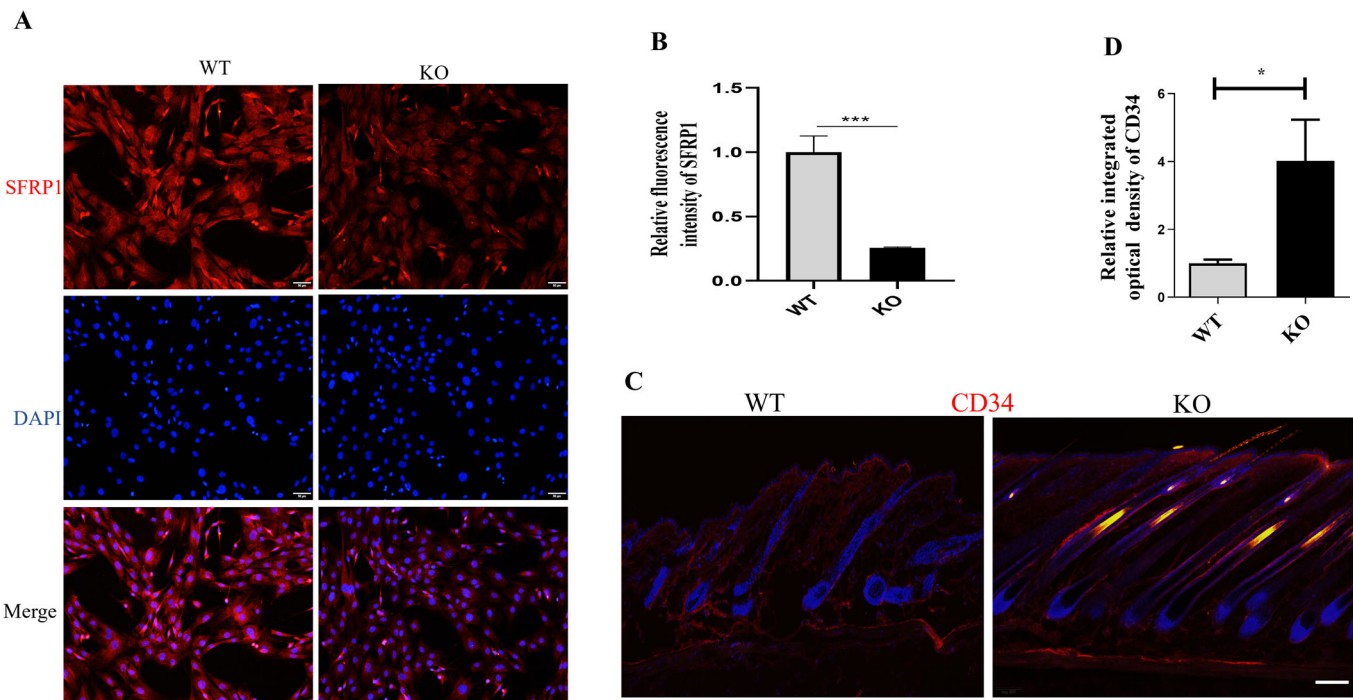

**Figure EV5.** *Gpr54* **deletion decreased SFRP1 in DPCs but increased CD34 in hair follicles.**

(A) The representative images of the immunofluorescence staining of SFRP1 in cultured *Gpr54* WT and KO DPCs. Scale bar = 50 μm. (B) The bar graph on the right represents the quantitative analysis of SFRP1 expression. Data are mean ± SD of three independent experiments. A paired *t*-test was performed, $N = 3$. ***$P = 0.0005$, WT versus KO. (C) *Gpr54* deletion increased the level of CD34 in hair follicle stem cells. The representative images of immunofluorescence staining of CD34 in the hair follicles of *Gpr54* WT and KO mice at day 25 after birth. Scale bar = 100 μm. (D) The bar graph represents the quantitative analysis for CD34 immunostaining in Fig. EV5C. Data are mean ± SD of three independent experiments. A paired *t*-test was performed, $N = 3$. *$P = 0.0128$, WT versus KO. Source data are available online for this figure.

