## [Peer Review File · EMBO Reports]

Gpr54 deletion accelerates hair cycle and hair regeneration

Weili Xia, Caibing Wang, Biao Guo, Zexin Tang, Xiyun Ye, and Yongyan Dang

Corresponding author(s): Yongyan Dang (yydang@bio.ecnu.edu.cn), Xiyun Ye (xyye@bio.ecnu.edu.cn)

Review Timeline:

Submission Date:	17th Apr 24
Editorial Decision:	14th May 24
Revision Received:	9th Aug 24
Editorial Decision:	24th Oct 24
Revision Received:	3rd Nov 24
Accepted:	13th Nov 24

Editor: Esther Schnapp

Transaction Report:

Dear Prof. Dang,

Thank you for the submission of your manuscript to EMBO reports. We have now received the comments from 2 referees that are pasted below.

As you will see, the referees acknowledge that the findings are potentially interesting. However, referee 1 points out that it remains to be demonstrated whether Gpr54 acts in the dermal papilla or in the HFSC themselves. This is an important concern that should be addressed. Both referees also raise several technical issues that need to be addressed. Please let me know if you have any comments or questions regarding the revisions and we can discuss this further, also in a video chat, if you like.

I would thus like to invite you to revise your manuscript with the understanding that the referee concerns must be fully addressed and their suggestions taken on board. Please address all referee concerns in a complete point-by-point response. Acceptance of the manuscript will depend on a positive outcome of a second round of review. It is EMBO reports policy to allow a single round of major revision only and acceptance or rejection of the manuscript will therefore depend on the completeness of your responses included in the next, final version of the manuscript.

We realize that it is difficult to revise to a specific deadline. In the interest of protecting the conceptual advance provided by the work, we recommend a revision within 3 months (14th Aug 2024). Please discuss the revision progress ahead of this time with the editor if you require more time to complete the revisions.

- 1) A data availability section providing access to data deposited in public databases is missing. If you have not deposited any data, please add a sentence to the data availability section that explains that.
- 2) Your manuscript contains statistics and error bars based on $n=2$. Please use scatter blots in these cases. No statistics should be calculated if $n=2$.

3) We replaced Supplementary Information with Expanded View (EV) Figures and Tables that are collapsible/expandable online. A maximum of 5 EV Figures can be typeset. EV Figures should be cited as 'Figure EV1, Figure EV2' etc... in the text and their respective legends should be included in the main text after the legends of regular figures.

5) a complete author checklist, which you can download from our author guidelines <https://www.embopress.org/page/journal/14693178/authorguide>. Please insert information in the checklist that is also reflected in the manuscript. The completed author checklist will also be part of the RPF.

6) Please note that all corresponding authors are required to supply an ORCID ID for their name upon submission of a revised manuscript (<https://orcid.org/>). Please find instructions on how to link your ORCID ID to your account in our manuscript tracking system in our Author guidelines <https://www.embopress.org/page/journal/14693178/authorguide#authorshipguidelines>

- the name of the statistical test used to generate error bars and P values,
- the number (n) of independent experiments (please specify technical or biological replicates) underlying each data point,
- the nature of the bars and error bars (s.d., s.e.m.),
- If the data are obtained from $n < 2$, use scatter blots showing the individual data points.

I look forward to seeing a revised form of your manuscript when it is ready.

Yours sincerely,

Referee #1:

Summary :

In this manuscript by Xia and Wang et al., the authors identified a novel axis of Gpr54-Nfatc3-SFRP1-Wnt signaling and attempted to demonstrate its role in regulation of quiescence of hair follicle stem cells (HFSCs) in vivo. Functions of Gpr54 and Nfatc3 isoform in HFSC biology were completely unknown before, and therefore, authors findings of involvement of Gpr54-Nfatc3-Sfrp1 in maintenance of HFSC quiescence advances our understanding of quiescence mechanisms used by adult stem cells such as HFSC. Authors used Gpr54 whole body knockout mice along with invitro cultures of dermal papillae cells (DPCs) and hair follicles to investigate effect of Gpr54 deletion on hair cycle and hair follicle regeneration. Through H&E analysis of skin tissue sections, they showed that Gpr54 KO mice hair follicles cycle faster than the WT mice. Further, authors also performed mechanistic study using DPC culture system and showed that the Gpr54 deletion leads to reduced level of a Wnt inhibitor SFRP1, and thereby activates Wnt signaling. As expected, increased Wnt signaling activity further led to faster hair cycling and accelerated hair follicle regeneration followed by depilation. In summary, authors revealed previously unknown functions of Gpr54 and Nfatc3, and demonstrated that they function through previously known SFRP1-Wnt signaling axis in controlling hair follicle cycling. However, I felt that some of the conclusions were not completely supported by the presented data and the manuscript lacks detailed description of the experiments and coherent connections between different sections. Importantly, the claim of precocious HFSC activation through signaling from DPCs, and not through HFSC autonomous effect is not completely convincing since author used whole body knockout mouse model. Below are my comments that may help authors to improve their manuscript.

Major comments :

1. In the entire study, the authors focused on investigating hair cycle promoting effect of Gpr54 deletion through dermal papilla cells (DPCs) and did not provide enough evidence or explanation for why Gpr54 deletion in HFSCs themselves is not sufficient to develop the observed phenotype. Importantly, all the components that authors studied here do very well express in HFSCs, including Grp54, Nfatc3, Sfrp1 (<https://doi.org/10.1073/pnas.1601599113>, DOI: 10.1126/science.1092436). Also, cell autonomous effect of Wnt signaling inhibitors have been well documented in activation of HFSCs and hair cycle progression. In fact, the authors own data shows that expression of NFATC3 in the hair follicle bulge compartment is stronger than in the dermal papilla in WT (Fig 4B), and therefore, there is a possibility that deletion of Gpr54 in HFSCs is sufficient to activate them precociously and develop the observed phenotype, and may not require abnormal signaling from dermal papillae cells. Ideally, authors need to employ conditional knockout mice to ablate Gpr54 only in HFSCs or DPC, but since it may not be feasible now, authors should at least provide thorough explanations and revise their conclusions wherever possible, including modifying the summary Fig 8.
2. Author's explanation of accelerated hair cycle doesn't match with the data presented in Fig 2. They mentioned that "Gpr54 KO mouse hair follicles quickly entered into the second anagen growth phase at P23 while WT mice remained in the first telogen at P25 in both male and female (Fig.2A)". However, images presented for P25 in Fig 2A clearly shows that WT follicles are in Ana III and not in the telogen. Also, C57BL6 mice are known to enter in anagen by P23. To demonstrate normal initiation of hair cycle in WT, as well as precocious activation of hair cycle in KO, authors need to perform comparative study of Ki67 staining in both WT and KO tissues at least at PD21, PD23 and PD25.
3. The authors have cultured the DPCs from vibrissae and have used them in all of their in vitro experiments (Based on the materials and methods). The authors cannot use the results from vibrissae DPCs to draw conclusions on back skin hair as both hair types are quite different in their physiology and cycling patterns. Further, α SMA is not known to express in DPCs. However, the authors show its expression in Fig 3D. This clearly shows that the cultured DPCs are possibly contaminated by other cell types like dermal sheath cells.
4. Authors should draw conclusions very carefully and consider revising some of their current conclusions. For instance, how does Cyclosporine mediated inhibition of Nfatc3 translocation into the nucleus of WT cells support the conclusion that Gpr54 regulates the activity of Nfatc3 through calcineurin? There is no involvement of Gpr54 in this inhibition experiment in Fig 4F.
5. As mentioned in Comment 1, Sfrp1 is expressed in HFSCs as well apart from DP. In fact, SFRP1 expression is higher in HFSCs in comparison to DP during telogen and loss of SFRP1 in HFSCs was previously shown to enhance HFSCs proliferation (Sunkara R. et al. Stem Cells, 2022). The authors should provide clear experimental evidence on how the loss of GRP54 only in DP affects HFSC proliferation through the loss of SFRP1 within DP or they should consider the possibility of precocious activation of HFSCs through the loss of Sfrp1 within them.
6. Did authors consider the same stage of hair follicle cycle of WT and KO mice for their FACS and cell cycle analysis in Fig 6? Or they used the same postnatal age? Again, there are no proper experimental details, so it's difficult to understand and interpret the data properly. If the authors had used the same postnatal day, then the changes in HFSC numbers and cell cycle profile would be obvious because HF of WT and KO would be different stages as KO hair follicles cycle faster (Fig 2), and therefore, stem cell numbers could be different because of different hair cycle stage, but not because of the effect of Gpr54 deletion. To conclude the effect of Gpr54 deletion on stem cell numbers, authors need to perform these analyses at the same stage of the hair follicle cycle of WT and KO mice, and not at the same age (PD).

Minor comments :

1. Differential expression of Gpr54 in different compartments of hair follicle is not clearly visible due to low resolution images in Fig 1A. Higher magnification images would help visualize different levels of Gpr54 in hair follicle bulge vs hair germ and dermal

papillae.

2. In Fig 1B, it is not clear from which cells the RNA was isolated. Is it from total skin or sorted DPCs or cultured DPCs? Please provide details of cell-type/tissue used in result section before concluding the results. Also, it's not clear how the stages were compared for calculating the significance in the graph, authors may consider using horizontal line above bars being compared, as they have done in some of their plots.
3. Again, there is no details of source and body part of human tissue used in Fig 1C. Also, tissue integrity seems to be compromised since there is no visible dermis region in the presented IF images.
4. Please transform images to show hair follicles in the same orientation in all images for better visualization. Scale bars are not visible properly.
5. Cartoon presented in supplementary Fig 1A to show comparative hair growth in WT and KO is not correct. WT mice are known to be in mid-anagen by P28, and authors own data presented in Fig 2A shows that WT mice are in anagen III by P25. However, the cartoon shows telogen until P28. Labelling for other stages requires corrections as well.
6. There is no explanation of transplantation assay in methods section. Did authors transplant shaved or depilated back skin at P21? Data presented in Fig 2C; D are after how many days of transplantation? The Y-axis of the relative length of HF in Fig 2D should be corrected as the HFs shown in Fig 2C do not appear to be 50mm/5cm.
7. Fig 2D citation is missing in the text.
8. Data presented in Figure 3D and 3E is not matching with their explanation in the text.
9. Fig 4B and 4F - scale bars are missing.
10. How did the authors measure calcineurin activity in Fig 4D?
11. Authors claim that "NFATc3 is abundantly distributed in the DP region of mouse hair follicles (Fig. 4B)". However, the expression of Nfatc3 in comparatively weaker in DP as compared to bulge. Author should consider revising the statement.
12. The IF staining pattern of SFRP1 in Fig 5E is not convincing. Shouldn't it be localized in cytoplasm and membrane, instead of in the nucleus?
13. Authors conclusion of increase hair follicle stem cells in KO mice through FACS analysis is not very well supported by their data. Authors need to calculate and compare the number of stem cells per follicle by IF staining to support their FACS data since the difference in stem cell numbers between WT and KO in their FACS data is marginal.
14. Check Y-axis title text of Fig 7D.
15. As mentioned in the major comments, there is a strong possibility of cell autonomous effect of Gpr54 deletion on HFSCs, so authors should consider revising their summary Fig 8 accordingly.

Referee #2:

In this manuscript, Xia et al investigate the role of Kisspeptin Receptor, GPR54, in homeostatic hair cycling as well as hair growth. First, using a GPR54 whole-body knockout mouse (GPR54-KO), they show that KO mice undergo a more rapid telogen-to-anagen transition following depilation. Transplants of mutant skin into nude mice were used to conclude that systemic effects (hormones, immune factors) do not underlie this phenotype. Using a combination of in vivo and in vitro approaches, the authors argue that GPR54 is particularly active in dermal papilla cells. Mechanistically, they claim that NFATc3 activity is regulated downstream of GPR54, with loss of GPR54 preventing NFATc3 from entering the nucleus to influence levels of the Wnt inhibitor SFRP1. Therefore, in GPR54-KO mice, the suppression of the Wnt pathway via the Frizzled receptor is ameliorated, potentially leading to enhanced paracrine signalling from dermal papilla cells to hair follicle stem cells. Finally, they show that the GPR54 inhibitor KP234 can increase hair growth rate in vitro as well as in vivo following depilation.

The central finding, that GPR54 can regulate rates of hair growth, is novel, and the efficacy of the KP234 inhibitor in modulating hair growth in vivo is intriguing. However, there are several significant issues with the data presented here that will need to be addressed before the paper is suitable for publication.

Major concerns:

1. The authors make several conclusions about the mechanism of action of GPR54 based on its staining pattern in the hair follicle and associated stroma. However, the staining in general is of low quality and appears variable between figures. It is essential that the authors validate the specificity of their GPR54 antibody by staining GPR54 KO skin tissue. This should be feasible given that the mutant mice are already on hand. Additionally, any specific claims about which cell populations express GPR54 should be accompanied by quantification of co-localization between GPR54 and specific markers for bulge, hair germ, etc (eg. what % of CD34+ and CD34- cells are GPR54 positive in Figure 6?)
2. The changes in GPR54 levels during hair follicle stages are hard to understand. In Figure 1A, the levels of GPR54 protein appear to be lower in general in every cell in the section. QPCR seems to have been performed on whole skin, of which hair follicle cells would only be a small proportion, suggesting that there could indeed be global changes. First: can the authors provide quantification for 1A to demonstrate that the lower signal in anagen is not just an artifact of staining or imaging conditions between slides? Second: how do the authors interpret these apparently very broad tissue-wide changes, given that they mainly focus on the role of GPR24 in DP cells?

3. The explant experiments are not sufficient to rule out effects from other cell types. The staining in Figure 1A indicates that GPR54 is expressed broadly in the stroma, which would be transplanted along with the follicles. The authors should soften their interpretation that the phenotype in the global knockout mice is due solely or mainly to effects in the follicle or DP. Also, histology and quantification of explanted hair follicles should be provided.

4. Along these lines, the authors should provide some evidence that the molecular mechanisms observed in vitro also occur in vivo. Are levels of versican and SMA increased in the DP cells of KO mice in vivo? Are levels of Wnt signaling (as indicated by nuclear Beta catenin, etc) increased in DP cells or hair follicle cells in vivo?

5. The authors propose that loss of GPR24 causes decreased calcineurin activation, which results in increased phosphorylation and nuclear sequestration of NFATc. Based on the results shown in vitro in Figure 4E, wouldn't immunostaining of GPR24 KO hair follicles be expected to show an altered subcellular distribution of NFATc3 protein as opposed to the almost complete lack of signal shown in Figure 4B? How do the authors explain this?

6. Quantification should be provided for the staining levels of SMP and versican in Figure 3C&D, and the Western blots in 3E and 5B and D.

7. How was KP234 administered to the mice? There is no information on this in the Methods.

8. The manuscript is severely under-referenced. Among others, statements in the introduction about the growth factors secreted by DP cells (paragraph 1), hair follicle cycling (paragraph 1), GPR54 expression patterns (paragraph 2), roles of GPR24 in puberty (paragraph 2) and mention of the Hair-GEL tool (paragraph 2) should all be referenced.

Minor concerns:

1. There are typos throughout, including in labels for many of the figures.

2. Information on the genetic background of the GPR24 KO mice should be provided in the methods.

Responses to reviewers

Dear reviewers,

Thank you for raising the insightful comments for our manuscript. We have performed additional experiments and included the new data in this revision. We have also carefully revised the manuscript text according to the suggestions. Below are point-to-point responses from us.

Table 1: Figure changes

Figures	Changes
Figure 1A	Improved the quality
Figure 1C	Deleted
Figure 5D	Deleted
Figure 8	Revised
Supplemental Figure 1A	Added
Supplemental Figure 1B	Added
Supplemental Figure 1A-C	Renumbered as Supplemental Figure 1C-E
Supplemental Figure 2A-D	Added
Supplemental Figure 2A-D 3A-D	Renumbered as Supplemental Figure
Supplemental Figure 4A-C	Added
Supplemental Figure 5A-C	Added

Referee #1:

In this manuscript by Xia and Wang et al., the authors identified a novel axis of Gpr54-Nfatc3-SFRP1-Wnt signaling and attempted to demonstrate its role in regulation of quiescence of hair follicle stem cells (HFSCs) in vivo. Functions of Gpr54 and Nfatc3 isoform in HFSC biology were completely unknown before, and therefore, authors findings of involvement of Gpr54-Nfatc3-Sfrp1 in maintenance of HFSC quiescence advances our understanding of quiescence mechanisms used by adult stem cells such as HFSC. Authors used Gpr54 whole body knockout mice along with invitro cultures of dermal papillae cells (DPCs) and hair follicles to investigate effect of Gpr54 deletion on hair cycle and hair follicle regeneration. Through H&E analysis of skin tissue sections, they showed that Gpr54 KO mice hair follicles cycle faster than the WT mice. Further, authors also performed mechanistic study using DPC culture system and

showed that the Gpr54 deletion leads to reduced level of a Wnt inhibitor SFRP1, and thereby activates Wnt signaling. As expected, increased Wnt signaling activity further led to faster hair cycling and accelerated hair follicle regeneration followed by depilation. In summary, authors revealed previously unknown functions of Gpr54 and Nfatc3, and demonstrated that they function through previously known SFRP1-Wnt signaling axis in controlling hair follicle cycling. However, I felt that some of the conclusions were not completely supported by the presented data and the manuscript lacks detailed description of the experiments and coherent connections between different sections. Importantly, the claim of precocious HFSC activation through signaling from DPCs, and not through HFSC autonomous effect is not completely convincing since author used whole body knockout mouse model. Below are my comments that may help authors to improve their manuscript.

Major comments :

1. In the entire study, the authors focused on investigating hair cycle promoting effect of Gpr54 deletion through dermal papilla cells (DPCs) and did not provide enough evidence or explanation for why Gpr54 deletion in HFSCs themselves is not sufficient to develop the observed phenotype. Importantly, all the components that authors studied here do very well express in HFSCs, including Grp54, Nfatc3, Sfrp1 (<https://doi.org/10.1073/pnas.1601599113>, DOI: 10.1126/science.1092436). Also, cell autonomous effect of Wnt signaling inhibitors have been well documented in activation of HFSCs and hair cycle progression. In fact, the authors own data shows that expression of NFATC3 in the hair follicle bulge compartment is stronger than in the dermal papilla in WT (Fig 4B), and therefore, there is a possibility that deletion of Gpr54 in HFSCs is sufficient to activate them precociously and develop the observed phenotype, and may not require abnormal signaling from dermal papillae cells. Ideally, authors need to employ conditional knockout mice to ablate Gpr54 only in HFSCs or DPC, but since it may not be feasible now, authors should at least provide thorough explanations and revise their conclusions wherever possible, including modifying the summary Fig 8.

Answer: It is a good suggestion. Indeed, the expression of NFATC3 in the hair follicle bulge compartment is stronger in WT hair follicles than the KO ones, indicating deletion of Gpr54 in HFSCs can also directly activate HFSCs. We had revised the related conclusion and modified Figure 8 according to the suggestion.

2. Author's explanation of accelerated hair cycle doesn't match with the data presented in Fig 2. They mentioned that "Gpr54 KO mouse hair follicles quickly entered into the second anagen growth phase at P23 while WT mice remained in the first telogen at P25 in both male and female (Fig.2A)". However, images presented for P25 in Fig 2A clearly show that WT follicles are in Ana III and not in the telogen. Also, C57BL6 mice are known to enter in anagen by P23. To demonstrate normal initiation of hair cycle in WT, as well as precocious activation of hair cycle in KO, authors need to perform comparative study of Ki67 staining in both WT and KO tissues at least at PD21, PD23 and PD25.

Answer: Thank you for the suggestion. We had changed "Gpr54 KO mouse hair follicles quickly entered into the second anagen growth phase at P23 while WT mice remained in the first telogen at P25 in both male and female (Fig.2A)" by " Gpr54 KO mouse hair follicles rapidly progressed to the second anagen VI phase by P25 while WT mice remained in the anagen III phase in both

males and females (Fig. 2A)." In addition, we conducted an immunofluorescence experiment to examine the differential expression of Ki67 in GPR54 WT and KO mice according to the suggestion. Consistently, the results indicated that, at 23 days, the number of Ki67-positive cells in the hair follicles of KO mice was higher than that in WT mice. At 25 days, the number of Ki67-positive cells in the hair bulb was significantly higher in KO mice than in WT mice (Supplemental Figure 1B).

3. The authors have cultured the DPCs from vibrissae and have used them in all of their in vitro experiments (Based on the materials and methods). The authors cannot use the results from vibrissae DPCs to draw conclusions on back skin hair as both hair types are quite different in their physiology and cycling patterns. Further, α SMA is not known to express in DPCs. However, the authors show its expression in Fig 3D. This clearly shows that the cultured DPCs are possibly contaminated by other cell types like dermal sheath cells.

Answer: Although the functions of vibrissae and back skin hair follicles differ, the cell types within these follicles, such as dermal papilla cells and matrix cells, exhibit similar functional characteristics and biological behaviors. This similarity allows the study of cellular processes in vibrissae follicles to provide insights into understanding other types of hair follicles. Additionally, we have conducted some studies on back skin hair follicles using immunofluorescence techniques (Figure 3,4 and supplemental Figure 1,2,4,5) to validate the results obtained from vibrissae follicles. They are consistent. Indeed, SMA is not known to be expressed in DPCs, whereas ALP and versican are recognized markers for DPC cells and can replace α SMA. Therefore, we have removed Figure 3D.

4. Authors should draw conclusions very carefully and consider revising some of their current conclusions. For instance, how does Cyclosporine mediated inhibition of Nfatc3 translocation into the nucleus of WT cells support the conclusion that Gpr54 regulates the activity of Nfatc3 through calcineurin? There is no involvement of Gpr54 in this inhibition experiment in Fig 4F.

Answer: We had deleted the conclusion that Gpr54 regulates the activity of Nfatc3 through calcineurin in Results according to the suggestion.

5. As mentioned in Comment 1, Sfrp1 is expressed in HFSCs as well apart from DP. In fact, SFRP1 expression is higher in HFSCs in comparison to DP during telogen and loss of SFRP1 in HFSCs was previously shown to enhance HFSCs proliferation (Sunkara R. et al. Stem Cells, 2022). The authors should provide clear experimental evidence on how the loss of GRP54 only in DP affects HFSC proliferation through the loss of SFRP1 within DP or they should consider the possibility of precocious activation of HFSCs through the loss of Sfrp1 within them.

Answer: It is a good suggestion. We found that the content of NFATC3 in HFSC cells was significantly lower in Gpr54 KO mice compared to WT mice, indicating Gpr54 deletion may directly activate HFSCs by inhibiting SFRP1. Therefore, the possibility cannot exclude that SFRP1 plays an important direct role in hair follicle stem cells. We have revised the relevant conclusions accordingly.

6. Did authors consider the same stage of hair follicle cycle of WT and KO mice for their FACS and cell cycle analysis in Fig 6? Or they used the same postnatal age? Again, there are no proper

experimental details, so it's difficult to understand and interpret the data properly. If the authors had used the same postnatal day, then the changes in HFSC numbers and cell cycle profile would be obvious because HFs of WT and KO would be different stages as KO hair follicles cycle faster (Fig 2), and therefore, stem cell numbers could be different because of different hair cycle stage, but not because of the effect of Gpr54 deletion. To conclude the effect of Gpr54 deletion on stem cell numbers, authors need to perform these analyses at the same stage of the hair follicle cycle of WT and KO mice, and not at the same age (PD).

Answer: As shown in Figure 2A, WT and KO mice are at the same hair cycle stage on postnatal day 19. To eliminate the interference of different hair cycles on the results, we thus chose to perform flow cytometric evaluation of the quantity of CD34+ and integrin $\alpha 6$ + cells in the hair follicles from Gpr54 WT and KO mice on postnatal day 19. Therefore, in Figures 6B and 6C, WT and KO mice are at the same hair cycle stage.

Minor comments :

1. Differential expression of Gpr54 in different compartments of hair follicle is not clearly visible due to low resolution images in Fig 1A. Higher magnification images would help visualize different levels of Gpr54 in hair follicle bulge vs hair germ and dermal papillae.

Answer: We had replaced the original images by higher magnification ones in Figure 1A.

2. In Fig 1B, it is not clear from which cells the RNA was isolated. Is it from total skin or sorted DPCs or cultured DPCs? Please provide details of cell-type/tissue used in result section before concluding the results. Also, it's not clear how the stages were compared for calculating the significance in the graph, authors may consider using horizontal line above bars being compared, as they have done in some of their plots.

Answer: In Figure 1B, the RNA was isolated from the total skin. We had added the horizontal line above bars being compared in Figure 1B according to the suggestions.

3. Again, there is no details of source and body part of human tissue used in Fig 1C. Also, tissue integrity seems to be compromised since there is no visible dermis region in the presented IF images.

Answer: It is a good suggestion. Indeed, we also found that the human hair follicle in Figure 1C is not very complete, so we have deleted the related results and conclusions.

4. Please transform images to show hair follicles in the same orientation in all images for better visualization. Scale bars are not visible properly.

Answer: We had transformed images to make hair follicles in the same orientation according to the suggestions.

5. Cartoon presented in supplementary Fig 1A to show comparative hair growth in WT and KO is not correct. WT mice are known to be in mid-anagen by P28, and authors own data presented in Fig 2A shows that WT mice are in anagen III by P25. However, the cartoon shows telogen until P28. Labelling for other stages requires corrections as well.

Answer: We had corrected the cartoon in the original supplementary Fig 1A (renumbered as supplementary Fig 1C) to make it consistent with the results of HE staining.

6. There is no explanation of transplantation assay in methods section. Did authors transplant shaved or depilated back skin at P21? Data presented in Fig 2C; D are after how many days of transplantation? The Y-axis of the relative length of HF in Fig 2D should be corrected as the HFs shown in Fig 2C do not appear to be 50mm/5cm.

Answer: Thank you for your suggestion. We had added the transplantation assay in methods section. The dorsal hair was shaved at P19. The hair growth was observed at 7 days after transplantation. We also corrected the Y-axis of the relative length of HF in Fig 2D.

7. Fig 2D citation is missing in the text.

Answer: We had added the citation of Figure 2D in the Results.

8. Data presented in Figure 3D and 3E is not matching with their explanation in the text.

Answer: We had corrected the citation for Figure 3D and 3E.

9. Fig 4B and 4F - scale bars are missing.

Answer: We had added the scale bars in Figure 4B and 4F

10. How did the authors measure calcineurin activity in Fig 4D?

Answer: We had added the details of calcineurin activity measurement in Materials and Methods.

11. Authors claim that "NFATc3 is abundantly distributed in the DP region of mouse hair follicles (Fig. 4B)". However, the expression of Nfatc3 in comparatively weaker in DP as compared to bulge. Author should consider revising the statement.

Answer: We had replaced the sentence "NFATc3 is abundantly distributed in the DP region of mouse hair follicles (Fig. 4B)" by "NFATc3 is abundantly distributed in the DP and bulge region of mouse hair follicles (Fig. 4B)".

12. The IF staining pattern of SFRP1 in Fig 5E is not convincing. Shouldn't it be localized in cytoplasm and membrane, instead of in the nucleus?

Answer: Indeed, SFRP1 is localized in both the cytoplasm and membrane in DPCs as shown in Figure 5E. Due to the small size of the images, it was difficult to observe this clearly. Therefore, we performed immunofluorescent staining of SFRP1 again to obtain high-magnification images (Supplemental Figure 5A). The results further confirmed that SFRP1 is localized in both the cytoplasm and membrane in DPCs.

13. Authors conclusion of increase hair follicle stem cells in KO mice through FACS analysis is not very well supported by their data. Authors need to calculate and compare the number of stem cells per follicle by IF staining to support their FACS data since the difference in stem cell numbers between WT and KO in their FACS data is marginal.

Answer: We added an immunofluorescence staining experiment for the stem cell marker CD34 as suggested. The results showed that CD34 expression levels were higher in knockout (KO) mice compared to wild-type (WT) mice (Supplemental Figure 5C), which is consistent with the flow cytometry (FACS) findings.

14. Check Y-axis title text of Fig 7D.

Answer: We had revised Y-axis title text of Fig 7D.

15. As mentioned in the major comments, there is a strong possibility of cell autonomous effect of Gpr54 deletion on HFSCs, so authors should consider revising their summary Fig 8 accordingly.

Answer: It is a good suggestion. We had corrected the summary for Figure 8.

Referee #2:

In this manuscript, Xia et al investigate the role of Kisspeptin Receptor, GPR54, in homeostatic hair cycling as well as hair growth. First, using a GPR54 whole-body knockout mouse (GPR54-KO), they show that KO mice undergo a more rapid telogen-to-anagen transition following depilation. Transplants of mutant skin into nude mice were used to conclude that systemic effects (hormones, immune factors) do not underlie this phenotype. Using a combination of in vivo and in vitro approaches, the authors argue that GRP54 is particularly active in dermal papilla cells. Mechanistically, they claim that NFATc3 activity is regulated downstream of GPR54, with loss of GRP54 preventing NFATc3 from entering the nucleus to influence levels of the Wnt inhibitor SFRP1. Therefore, in GPR54-KO mice, the suppression of the Wnt pathway via the Frizzled receptor is ameliorated, potentially leading to enhanced paracrine signalling from dermal papilla cells to hair follicle stem cells. Finally, they show that the GPR54 inhibitor KP234 can increase hair growth rate in vitro as well as in vivo following depilation.

The central finding, that GPR54 can regulate rates of hair growth, is novel, and the efficacy of the KP234 inhibitor in modulating hair growth in vivo is intriguing. However, there are several significant issues with the data presented here that will need to be addressed before the paper is suitable for publication.

Major concerns:

1. The authors make several conclusions about the mechanism of action of GPR54 based on its staining pattern in the hair follicle and associated stroma. However, the staining in general is of low quality and appears variable between figures. It is essential that the authors validate the specificity of their GPR54 antibody by staining GPR54 KO skin tissue. This should be feasible given that the mutant mice are already on hand. Additionally, any specific claims about which cell populations express GPR54 should be accompanied by quantification of co-localization between GPR54 and specific markers for bulge, hair germ, etc (eg. what % of CD34+ and CD34- cells are GPR54 positive in Figure 6?)

Answer: We had added the results of immunofluorescent staining in GPR54 KO skin tissue and validate that the specificity of GPR54 antibody is OK (Supplemental Figure 1A). Due to the directional nature of hair follicles, it is challenging to obtain completely consistent hair follicles in sections of mouse skin. Some structures of certain hair follicles may not be fully visible. Therefore, it is difficult for us to precisely make the quantitative analysis data for the colocalization of Gpr54 and CD34.

2. The changes in GPR54 levels during hair follicle stages are hard to understand. In Figure 1A,

the levels of GPR54 protein appear to be lower in general in every cell in the section. QPCR seems to have been performed on whole skin, of which hair follicle cells would only be a small proportion, suggesting that there could indeed be global changes. First: can the authors provide quantification for 1A to demonstrate that the lower signal in anagen is not just an artifact of staining or imaging conditions between slides? Second: how do the authors interpret these apparently very broad tissue-wide changes, given that they mainly focus on the role of GPR24 in DP cells?

Answer: Thank you for the suggestion. We apologize for the difficulty in conducting precise quantitative analysis due to the varying morphology of each hair follicle in the images. Indeed, we found that GPR54 is distributed in both the DP and the bulge regions. Although GPR54 plays an important role in DPC cells, its role in HFSCs cannot be ignored or excluded, so we have revised the related conclusions and Figure 8 as well as its summary in this manuscript.

3. The explant experiments are not sufficient to rule out effects from other cell types. The staining in Figure 1A indicates that GPR54 is expressed broadly in the stroma, which would be transplanted along with the follicles. The authors should soften their interpretation that the phenotype in the global knockout mice is due solely or mainly to effects in the follicle or DP. Also, histology and quantification of explanted hair follicles should be provided.

Answer: It is a good suggestion. We had revised conclusion that the phenotype in the global knockout mice is mainly to effects in the follicle or DP. Moreover, we had added the results of histology and quantification of explanted hair follicles (Supplemental Figure 2A and 2B).

4. Along these lines, the authors should provide some evidence that the molecular mechanisms observed *in vitro* also occur *in vivo*. Are levels of versican and SMA increased in the DP cells of KO mice *in vivo*? Are levels of Wnt signaling (as indicated by nuclear Beta catenin, etc) increased in DP cells or hair follicle cells *in vivo*?

Answer: We had added the results of versican increased in the DP cells of KO mice *in vivo* (Supplemental Figure 2D). Considering SMA is not known to be expressed in DPCs while ALP is a recognized marker for DPC cells and can replace α SMA. Therefore, we have removed Figure 3D. Moreover, we also increased the results of β -catenin in hair follicle cells *in vivo* (Supplemental Figure 4C).

5. The authors propose that loss of GPR54 causes decreased calcineurin activation, which results in increased phosphorylation and nuclear sequestration of NFATc. Based on the results shown *in vitro* in Figure 4E, wouldn't immunostaining of GPR54 KO hair follicles be expected to show an altered subcellular distribution of NFATc3 protein as opposed to the almost complete lack of signal shown in Figure 4B? How do the authors explain this?

Answer: Due to the changes in the hair cycle phases of anagen, catagen, and telogen, the state of DP cells may vary at different time. We speculate that NFATc3 is a short-lived protein and the NFATc3 phosphorylation (P-NFATc3) may promote its degradation, which could explain the faint staining observed in Figure 4B. We have included a description of this possibility in the discussion of our manuscript.

6. Quantification should be provided for the staining levels of SMP and versican in Figure 3C&D,

and the Western blots in 5B and D.

Answer: We had added the quantifications for the staining levels of ALP and versican in Figure 3B and 3C (Supplemental Figure 2C). The results of SMP staining were removed according to the suggestions. In addition, we also added the quantifications for the Western blots in 5B and 5D (Supplemental Figure 4A and 4B).

7. How was KP234 administered to the mice? There is no information on this in the Methods.

Answer: KP234 was administered subcutaneously. We had added the information in Methods.

8. The manuscript is severely under-referenced. Among others, statements in the introduction about the growth factors secreted by DP cells (paragraph 1), hair follicle cycling (paragraph 1), GPR54 expression patterns (paragraph 2), roles of GPR24 in puberty (paragraph 2) and mention of the Hair-GEL tool (paragraph 2) should all be referenced.

Answer: We had added the references according to the suggestions.

Minor concerns:

1. There are typos throughout, including in labels for many of the figures.

Answer: We had carefully revised the type errors in all this manuscript.

2. Information on the genetic background of the GPR54 KO mice should be provided in the methods.

Answer: We had added the background of the GPR54 KO mice in the methods.

Best regards

Sincerely

Yongyan Dang

Image Revision Explanation

Dear Editor,

We sincerely apologize for the error that occurred during the image compilation process due to the large number of HE images, with different authors handling various images. As a result, a mistake was made during the final integration. We have now replaced the misused skin hair follicle image of WT mice in the catagen phase in Figure S.1D with a new image (the file date is the same as that of the KO mice image) in Page 9.

We deeply regret any inconvenience this may have caused. We would greatly appreciate your understanding in this matter.

Thank you very much for your patience.

Sincerely,

Yongyan Dang, PhD

Dear Prof. Dang,

Thank you for the submission of your revised manuscript. We have now received the enclosed reports from the referees. Both referees still have a few more suggestions that I would like you to address and incorporate before we can proceed with the official acceptance of your manuscript. Please co-submit a ms with track changes to ease its assessment and a detailed point-by-point response.

A few editorial requests will also need to be addressed:

- Please add a DATA AVAILABILITY SECTION (DAS) to the end of the Methods. The DAS should list access to data generated in this study and deposited in public databases. If you have not deposited any data, please add a sentence to the DAS that explains that. The question about the DAS in the author checklist also needs to be answered.
 - Please remove the author credits from the ms file. All credits need to be entered during online ms submission.
 - Please correct the conflict of interest subheading to "Disclosure and Competing Interest Statement".
 - The supplemental table 1 can be part of the Reagents and Tools table that needs to be added to the ms files. This table should list key reagents, experimental models, software and relevant equipment and including their sources and relevant identifiers. A downloadable templates (.docx) for the Reagents and Tools Table can be found in our author guidelines: <<https://www.embopress.org/page/journal/14693178/authorguide#manuscriptpreparation>>
 - The suppl. figure legends are in fact EV figure legends and need to be removed from the current file and added to the manuscript file, after the main figure legends and under the heading "Expanded View Figure Legends". The nomenclature and ms callouts should be corrected to "Figure EV1" - "Figure EV5".
 - For the main figures, source data (SD) need to be uploaded as 1 (zipped) folder per main figure. The SD checklist needs to be completed. The SD for all EV figures need to be zipped into one single folder.
 - There is a callout for Fig 3E but that panel is not in the figure or legend.
 - The correct order of the manuscript's section is: Abstract / Introduction / Results / Discussion / Methods / Acknowledgements / Disclosure and competing interests statement / References / Figure legends / Expanded View Figure legends
1. Please note that figure 5c does not describe the bar graph in the corresponding legend. This needs to be rectified.
 2. Please list the exact p values in the legends of figures 1b; 2e; 3d; 4a, c-d, f; 5a, c, e; 6b-d; 7b, d, supplementary figures 2c; 3d; 4a, c; 5b as reasonable.
 3. Although 'n' is provided, please describe the nature of entity for 'n' in the legend of figure 7b.
- Please re-write the abstract and avoid all overstatements, as also pointed out by the referees. Please also explain what DPC is in the abstract. The abstract further needs to be written in present tense.

EMBO press papers are accompanied online by A) a short (1-2 sentences) summary of the findings and their significance, B) 2-3 bullet points highlighting key results and C) a synopsis image that is exactly 550 pixels wide and 200-600 pixels high (the height is variable). The synopsis image should provide a sketch of the major findings, like a graphical abstract. Please note that text needs to be readable at the final size. Please send us this information along with the final manuscript.

Referee #1:

In this revised manuscript by Xia and Wang et al., the authors have improved the manuscript by addressing my concerns. However, there are still some questions/concerns that need to be addressed to clearly conclude the key findings of the study.

1) In response to my major concern of cell-autonomous effect of Gpr54 deletion in HFSC, the authors have agreed by saying "It is a good suggestion. Indeed, the expression of NFATC3 in the hair follicle bulge compartment is stronger in WT hair follicles than the KO ones, indicating deletion of Gpr54 in HFSCs can also directly activate HFSCs. We had revised the related conclusion and modified Figure 8 according to the suggestion." However, in the main text of the manuscript, the effect of Gpr54 KO on hair cycle is still majorly explained through DPC and not through HFSCs, including in the subtitles of the result section.

2) Again, in the response to reviewers' comments, the authors have said that "We had deleted the conclusion that Gpr54 regulates the activity of Nfatc3 through calcineurin in Results according to the suggestion." However, from the title to concluding statement in discussion, authors are still describing their result through calcineurin-NFATC3-SFRP1-Wnt- β -catenin signaling pathway.

3) Though it is shown that hair follicles of GPR54 KO mice enter into anagen faster compared to WT, it is still not clear from Fig.2 why they enter into catagen and telogen at similar times. Further, as the KO mice undergo more frequent hair cycles compared to WT, is there a loss in the stem cell pool over time? Authors should provide explanations of these points in discussion.

4) I still think that concise and careful writing would help readers understand the paper easily. For instance, what should readers understand from this sentence - "The brightest GPR54 fluorescence intensity in DP and HFSCs implies its most significant expression in DPCs (Fig.1A)." There are also few typos, e.g., Gpr45 instead of Gpr54.

5) The follicle shown in the model Fig.8 is in anagen. The authors should replace it with a telogen follicle as the entire study is about the effect of GPR54 KO on telogen to anagen transition.

6) In the response to reviewers' comments, authors said that they have addressed the concerns, but it's not reflecting in the revised manuscript satisfactorily.

Referee #2:

In their revision, the authors had addressed some, but not all, of my concerns. I appreciate that quantification of fluorescent signal in hair follicle sections may be difficult, but many of the in vivo claims are based on this staining, so reproducibility and consistency are important.

Since the only quantification of GPR54 levels in the manuscript come from QPCR data from whole skin (Figure 1B), the authors should make it clear that this analysis comes from bulk skin and may not accurately reflect the situation in hair follicles specifically.

The authors have provided additional images to demonstrate that Bcatenin levels are up in KO mice (Figure S4B). However the image provided for the control tissue is clearly out of focus, so it seems impossible to draw a proper comparison. A higher quality image should be provided, and a zoomed in region should be shown to demonstrate whether any of the Bcatenin localizes to the nucleus in either genotype.

I appreciate the hypothesis that phosphorylated NFATc3 is less stable, explaining the lack of signal in the mutant follicles. But this seems inconsistent with the higher levels of p-NFATc3 detected by Western blot in Figure 4E. This should be clarified in the discussion.

Responses to reviewers

Dear reviewers,

Thank you for raising the insightful comments for our manuscript. We have also carefully revised the manuscript text according to the suggestions. Below are point-to-point responses from us.

Referee #1:

In this revised manuscript by Xia and Wang et al., the authors have improved the manuscript by addressing my concerns. However, there are still some questions/concerns that need to be addressed to clearly conclude the key findings of the study.

1) In response to my major concern of cell-autonomous effect of Gpr54 deletion in HFSC, the authors have agreed by saying "It is a good suggestion. Indeed, the expression of NFATC3 in the hair follicle bulge compartment is stronger in WT hair follicles than the KO ones, indicating deletion of Gpr54 in HFSCs can also directly activate HFSCs. We had revised the related conclusion and modified Figure 8 according to the suggestion." However, in the main text of the manuscript, the effect of Gpr54 KO on hair cycle is still majorly explained through DPC and not through HFSCs, including in the subtitles of the result section.

Answer : Thank you for the suggestions. We had revised the inaccurate statement regarding the effect of Gpr54 KO on the hair cycle primarily through DPC rather than HFSCs, in the results and discussion sections, including the relevant subtitles.

2) Again, in the response to reviewers' comments, the authors have said that "We had deleted the conclusion that Gpr54 regulates the activity of Nfatc3 through calcineurin in Results according to the suggestion." However, from the title to concluding statement in discussion, authors are still describing their result through calcineurin-NFATC3-SFRP1-Wnt- β -catenin signaling pathway.

Answer : We have revised the statements in the manuscript regarding Gpr54's regulation of Nfatc3 activity through calcineurin in accordance with the suggestion.

3) Though it is shown that hair follicles of GPR54 KO mice enter into anagen faster compared to WT, it is still not clear from Fig.2 why they enter into catagen and telogen at similar times. Further, as the KO mice undergo more frequent hair cycles compared to WT, is there a loss in the stem cell pool over time? Authors should provide explanations of these points in discussion.

Answer : I'm very sorry; we may not have described it clearly. This time, we added a detailed description of the differences between WT and KO mice in the results section to better illustrate these differences. Moreover , we added an analysis in the Discussion section addressing the question: "As the KO mice undergo more frequent hair cycles compared to WT, is there a loss in the stem cell pool over time?"

4) I still think that concise and careful writing would help readers understand the paper easily. For instance, what should readers understand from this sentence - "The brightest GPR54 fluorescence intensity in DP and HFSCs implies its most significant expression in DPCs (Fig.1A)." There are also few typos, e.g., Gpr45 instead of Gpr54.

Answer :Thank you for the suggestions. We had carefully revised the writing and spelling errors in the manuscript.

5) The follicle shown in the model Fig.8 is in anagen. The authors should replace it with a telogen follicle as the entire study is about the effect of GPR54 KO on telogen to anagen transition.

Answer : It is a good suggestion. We have revised Figure 8, replacing the anagen hair follicle in the model with a telogen hair follicle.

6) In the response to reviewers' comments, authors said that they have addressed the concerns, but it's not reflecting in the revised manuscript satisfactorily.

Answer : We appreciate your feedback and apologize for any oversight. We have carefully reviewed the manuscript again and made further revisions to ensure that the concerns are fully addressed and clearly reflected in the text.

Referee #2:

1. In their revision, the authors had addressed some, but not all, of my concerns. I appreciate that quantification of fluorescent signal in hair follicle sections may be difficult, but many of the in vivo claims are based on this staining, so reproducibility and consistency are important.

Answer : Thank you for the suggestions. We conducted quantitative analysis of fluorescence results at the hair follicle level in Figure 4B and Supplemental Figures 1B, 2D, 4B, and 5C. However, due to significant structural differences in hair follicles across different stages (anagen, catagen, and telogen), cross-group comparisons were difficult. Therefore, it was challenging for us to perform quantitative analysis on the in vivo fluorescence images such as Figures 1A and 3A, which show immunofluorescent staining of wild-type mice at various hair cycle stages. We have revised the relevant descriptions in the results section to avoid confusion and enhance clarity.

2. Since the only quantification of GPR54 levels in the manuscript come from QPCR data from whole skin (Figure 1B), the authors should make it clear that this analysis comes from bulk skin and may not accurately reflect the situation in hair follicles specifically.

Answer : It is a good suggestion. We have added the detailed descriptions in the Result sections.

3. The authors have provided additional images to demonstrate that β -catenin levels are up in KO mice (Figure S4B). However the image provided for the control tissue is clearly out of focus, so it seems impossible to draw a proper comparison. A higher quality image should be provided, and a zoomed in region should be shown to demonstrate whether any of the Bcatenin localizes to the nucleus in either genotype.

Answer : We have updated the control image with new ones and partially enlarged the images according to the suggestions. Additionally, we conducted an

analysis of the number of β -catenin-positive cells in the nucleus (as seen in Supplemental Figure 4C).

4. I appreciate the hypothesis that phosphorylated NFATc3 is less stable, explaining the lack of signal in the mutant follicles. But this seems inconsistent with the higher levels of p-NFATc3 detected by Western blot in Figure 4E. This should be clarified in the discussion.

Answer : It is a good suggestion. Thank you for the suggestion. We speculated that the higher level of p-NFATC3 in DPC cells of KO mice compared to WT mice, as shown in Figure 4E, may result from Western blot capturing phosphorylation at a specific time point, allowing detection of the phosphorylated protein before it is fully degraded. We have added this explanation to the Discussion section.

Best regards

Sincerely

Yongyan Dang, PhD

Prof. Yongyan Dang
Shanghai Key Laboratory of Regulatory Biology, Institute of Biomedical Sciences, School of Life Sciences, East China Normal University
China

Dear Prof. Dang,

I am very pleased to accept your manuscript for publication in the next available issue of EMBO reports. Thank you for your contribution to our journal.

Yours sincerely,
